# All-atom molecular dynamics of the HBV capsid reveals insights into biological function and cryo-EM resolution limits

**Jodi A Hadden[1]\*, Juan R Perilla[1], Christopher John Schlicksup[2], Balasubramanian Venkatakrishnan[2], Adam Zlotnick[2], Klaus Schulten[3,4]†**

[1]Department of Chemistry and Biochemistry, University of Delaware, Newark, United States; [2]Department of Molecular and Cellular Biochemistry, Indiana University, Bloomington, United States; [3]Department of Physics, University of Illinois at Urbana-Champaign, Urbana, United States; [4]Beckman Institute, University of Illinois at Urbana-Champaign, Urbana, United States

**Abstract** The hepatitis B virus capsid represents a promising therapeutic target. Experiments suggest the capsid must be flexible to function; however, capsid structure and dynamics have not been thoroughly characterized in the absence of icosahedral symmetry constraints. Here, all-atom molecular dynamics simulations are leveraged to investigate the capsid without symmetry bias, enabling study of capsid flexibility and its implications for biological function and cryo-EM resolution limits. Simulation results confirm flexibility and reveal a propensity for asymmetric distortion. The capsid's influence on ionic species suggests a mechanism for modulating the display of cellular signals and implicates the capsid's triangular pores as the location of signal exposure. A theoretical image reconstruction performed using simulated conformations indicates how capsid flexibility may limit the resolution of cryo-EM. Overall, the present work provides functional insight beyond what is accessible to experimental methods and raises important considerations regarding asymmetry in structural studies of icosahedral virus capsids.

DOI: https://doi.org/10.7554/eLife.32478.001

**\*For correspondence:**
jhadden@udel.edu

†In memoriam

## Introduction

Hepatitis B virus (HBV) is a spherical pararetrovirus that infects hepatocytes. Despite the availability of a vaccine, around 240 people worldwide suffer from chronic HBV infection (*Ott et al., 2012*), placing them at risk for severe liver disease, such as cirrhosis and cancer. Among new strategies being explored for the treatment of HBV, disruption of its capsid represents a promising approach (*Zlotnick and Mukhopadhyay, 2011*).

The HBV capsid typically manifests as a T = 4 icosahedral particle, 36 nm in diameter, composed of 120 copies of capsid protein (Cp) homodimer. The capsid carries out important functional roles throughout the HBV infection cycle, assembling to enclose the viral genome (pregenomic RNA, pgRNA), serving as a container for reverse transcription of single-stranded linear pgRNA to circular double-stranded DNA (ds-DNA), displaying signals for intracellular trafficking and the binding of viral envelope glycoproteins, and dissociating to release its cargo to the host cell nucleus (*Venkatakrishnan and Zlotnick, 2016*). The Cp assembly domain (Cp149) represents the minimum protein length required to achieve self-association of dimers into archetypal capsids (*Zlotnick et al., 1996*). Atomic models of the assembly domain capsid (4 MDa) have been determined crystallographically via symmetry averaging (*Wynne et al., 1999*; *Bourne et al., 2006*; *Katen et al., 2013*; *Venkatakrishnan et al., 2016*). Full-length Cp includes 34 additional residues on its C-terminus (Cp183). The C-terminal domain (CTD) tails interact with pgRNA in the immature capsid, extrude to

the capsid exterior to signal key events during the viral life cycle, and are subject to phosphorylation and dephosphorylation (*Venkatakrishnan et al., 2016*).

Development of therapeutic interventions that target the HBV capsid can be supported by all-atom characterization of its structure, dynamics, and biophysical properties. However, the size and complexity of the complete capsid have rendered it difficult to study without enforcing assumptions based on icosahedral symmetry. For decades, researchers have relied on symmetry to obtain experimental structures of icosahedral virus capsids, as well as to make computational predictions regarding their dynamical behavior. Unfortunately, the results generated by such approaches may provide an incomplete and/or oversimplified description of a complex molecular machine, thus, limiting researchers' ability to ascertain the mechanisms by which it functions.

The present work addresses the need for all-atom characterization of the HBV capsid without symmetry bias using molecular dynamics (MD) simulations on the microsecond timescale. MD simulations have emerged as a powerful technique to model and investigate large biomolecules (*Goh et al., 2016*; *Perilla et al., 2015*), and advances in supercomputing now enable their application to complete virus capsids and virus-like particles at full chemical detail (*Freddolino et al., 2006*; *Larsson et al., 2012*; *Andoh et al., 2014*; *Tarasova et al., 2017a*; *Tarasova et al., 2017b*; *Perilla and Schulten, 2017*). The HBV capsid has been examined previously with coarse-grained simulations (*Arkhipov et al., 2009*) and in preliminary work that encompassed a timescale of only 0.1 μs (*Perilla et al., 2016*). Here, computational study of the HBV capsid yields an all-atom description of unconstrained dynamics based on a timescale an order of magnitude greater than hitherto explored and represents the most extensive unbiased simulation achieved to date for a T > 1 icosahedral virus capsid. The availability of an unsymmetrized conformational ensemble sampled over 1 μs provides the unique opportunity to study intrinsic capsid flexibility, its potential role in viral function, and its relationship to resolution obtained with single-particle image reconstruction, a technique used widely in structure determination by cryo-electron microscopy (cryo-EM).

Simulation results reveal remarkable asymmetry in capsid shape and essential dynamics, supporting hypotheses that the ability to accommodate asymmetric distortion plays a key role in its biological function. Findings further demonstrate the capsid's influence on its surrounding solvent environment, particularly with respect to the behavior of charged species, suggesting a mechanism to modulate the display of nuclear localization signals. Observations implicate the capsid's triangular pores as participating in the cellular signaling process. The identification of sodium localization within trimer-of-dimer interfaces could contribute to accelerated capsid assembly under high salt concentrations. Finally, a theoretical single-particle image reconstruction performed using capsid conformations sampled during simulation indicates that capsid flexibility may represent a major limiting factor to achieving true atomic (1–2 Å) resolution with cryo-EM.

## Results

### Capsid simulation reaches equilibrium after 0.1 μs

An all-atom model of the HBV capsid assembly domain (Cp149, *Figure 1a*) was constructed based on an available apo-form crystal structure (PDB ID 2G33, 3.96 Å) (*Bourne et al., 2006*) and simulated without restraints for a total of 1.1 μs. Including explicit water molecules and 150 mM NaCl, the simulation encompassed nearly 6 M atoms. Assessment of the capsid's size, stability, and morphology demonstrates that 0.1 μs of simulation time was required for the capsid to relax from its crystallographic state and reach structural equilibrium under physiological conditions.

The inner volume of the capsid was estimated by fitting a 240-face polyhedron to its symmetry axes (*Figure 1b*, described in Materials and methods). Following the release of simulation restraints, volume increased for 0.05 μs (*Figure 1c*). An overall expansion of capsid volume by 4% suggests that the crystal structure was substantially affected by crystal contacts. The mutual similarity of capsid conformers sampled during simulation was assessed by measuring Cα root-mean-square-deviation (RMSD) pairwise between them. RMSD is defined as the root-mean-square-average distance between two optimally superimposed biomolecular structures. Cα RMSD between capsid conformers converged to within 5 Å around 0.1 μs (*Figure 1—figure supplement 1*), indicating the equilibration time required for the model to reach a stable configuration. Interestingly, the all-atom

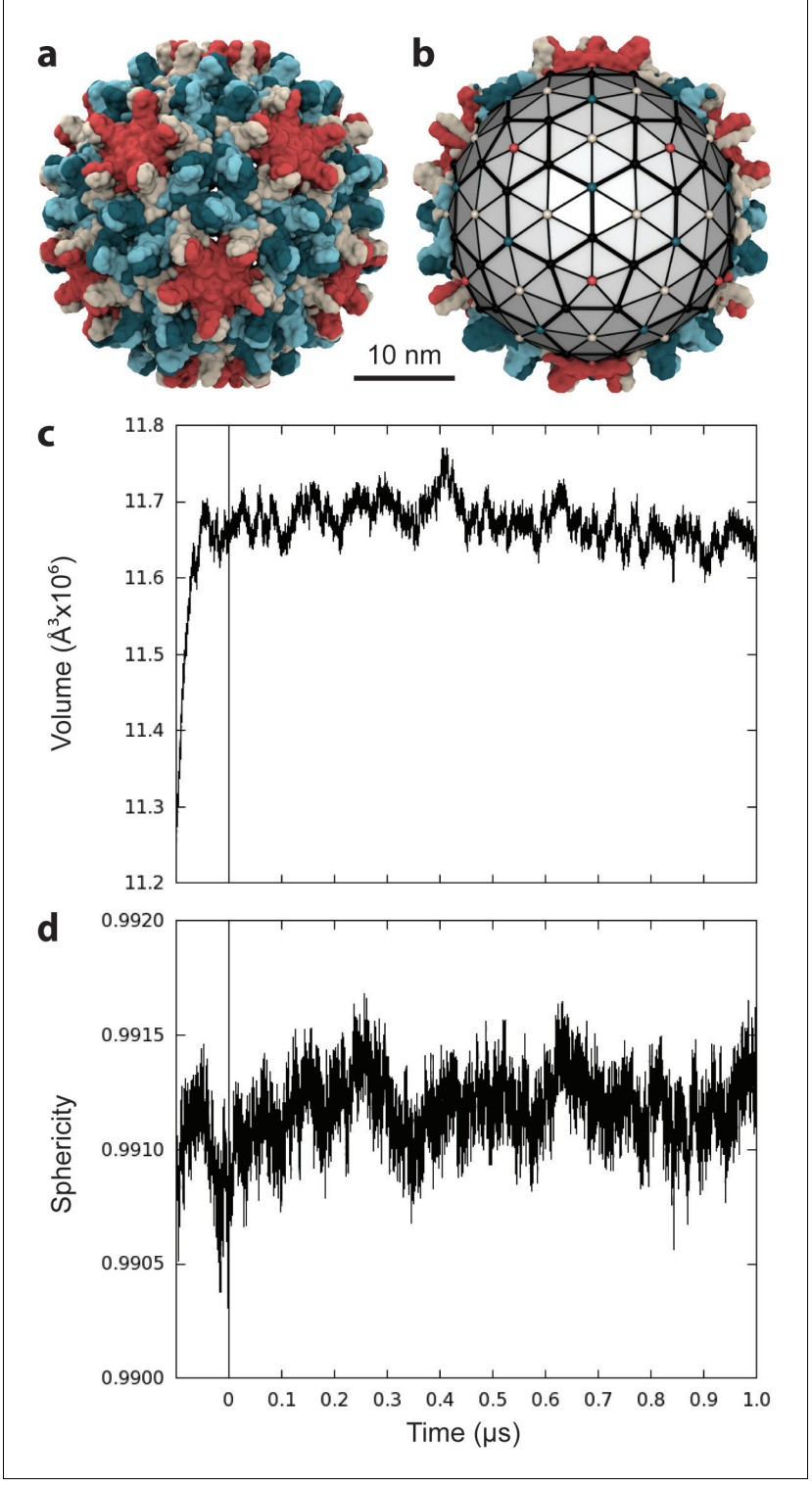

**Figure 1.** Capsid size, stability, and morphology. (**a**) The T = 4 HBV capsid is composed of 120 copies of Cp homodimer, where A (red) and B (beige) quasi-equivalent chains comprise AB dimers, and C (cyan) and D (blue) quasi-equivalent chains comprise CD dimers. (**b**) A polyhedron of 240 triangular faces was used to estimate capsid inner volume and sphericity; see Materials and methods. Polyhedron shown within a cross-section of the capsid, with fivefold (red spheres), sixfold (beige spheres), and threefold (blue spheres) vertices indicated. (**c**) Capsid inner volume increased and stabilized within 0.05 μs. (**d**) Capsid sphericity remained high throughout the simulation.
DOI: https://doi.org/10.7554/eLife.32478.002

*Figure 1 continued on next page*

*Figure 1 continued*

The following source data and figure supplement are available for figure 1:

**Source data 1.** Raw data for plots 1c-d in plain text format.
DOI: https://doi.org/10.7554/eLife.32478.004
**Figure supplement 1.** RMSD of complete capsid.
DOI: https://doi.org/10.7554/eLife.32478.003

model of the human immunodeficiency virus type 1 (HIV-1) capsid (*Zhao et al., 2013*) was observed to shrink during simulation and reached equilibrium only after 0.4 µs (*Perilla and Schulten, 2017*).

The sphericity of the HBV capsid, a measure of how closely its 3D shape approaches that of a sphere (*Wadell, 1935*), was also estimated based on the 240-face polyhedron (*Figure 1b*). A perfect sphere has unit sphericity, while a regular icosahedron has sphericity ~0.939. The capsid's morphology remained highly spherical throughout the course of simulation, fluctuating within 0.15% of the value calculated for the crystal structure (*Figure 1d*). While fluctuations in sphericity clearly arise from fluctuations in capsid shape, decreased sphericity does not necessarily indicate asymmetry.

The 0.1 µs equilibration time required for the HBV capsid model is consistent with previous all-atom simulation work on the similarly sized poliovirus capsid, which attempted to correct for anticipated volume expansion in the initial model, prior to simulation (*Andoh et al., 2014*). Equilibration times are expected to be even longer for larger capsids (*Perilla and Schulten, 2017*) and capsids whose experimental structures were determined under conditions very different from their native environment. Given that most other all-atom investigations of virus capsids have reached timescales of only tens of nanoseconds, these observations underscore the likely necessity of future capsid simulation studies to sample temporal regimes of hundreds of nanoseconds in order to produce relaxed conformational ensembles conducive to the measurement of equilibrium biophysical properties. The final 1 µs of simulation obtained in the present work describes a relaxed and structurally stable model of the HBV capsid that was employed for further analysis. An animation depicting capsid dynamics over this window is provided in *Video 1*.

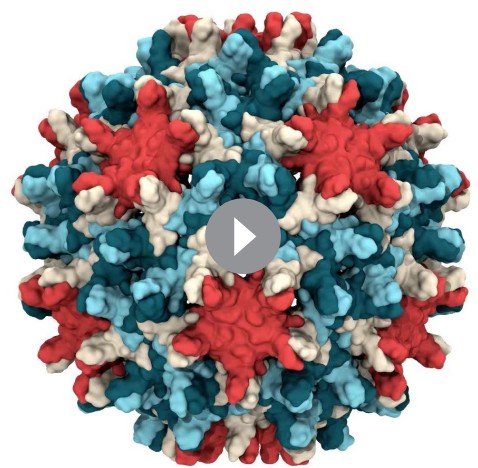

**Video 1.** All-atom MD simulation of the HBV capsid. Animation illustrating dynamics of the HBV capsid sampled over 1 µs of all-atom MD simulation. The capsid is composed of 120 copies of Cp homodimer, where Cp occupies quasi-equivalent chain positions A (red), B (beige), C (cyan), and D (blue). Animation rendered using VMD.
DOI: https://doi.org/10.7554/eLife.32478.005

## Local capsid dynamics depend on quasi-equivalence

Above all else, simulation of the HBV capsid reveals that it is remarkably flexible. Local capsid dynamics arise from the internal motions of constituent Cp dimers (*Figure 2a*). The flexibility of Cp is likely an essential feature, as dimers appear to be capable of a conformational switch that facilitates assembly (*Packianathan et al., 2010*), and assembled capsids can accommodate two different T numbers (*Venkatakrishnan and Zlotnick, 2016*).

Cp flexibility was assessed by measuring Cα root-mean-square-fluctuations (RMSF) for dimers extracted from the capsid simulation. RMSF is defined as the root mean-square-average distance between an atom and its average position within an ensemble of biomolecular structures. Cα RMSF profiles indicate that the C-terminal extended structure beyond helix 5 is the most mobile segment of Cp (*Figure 2b–c*). Values are 9.8 ± 3.1 Å at V149-Cα and 2.1 ± 0.7 Å at T142-Cα , the last crystallographically ordered residue in most experimental structures (*Wynne et al., 1999*; *Bourne et al., 2006*; *Katen et al., 2013*; *Venkatakrishnan et al., 2016*). In Cp183,

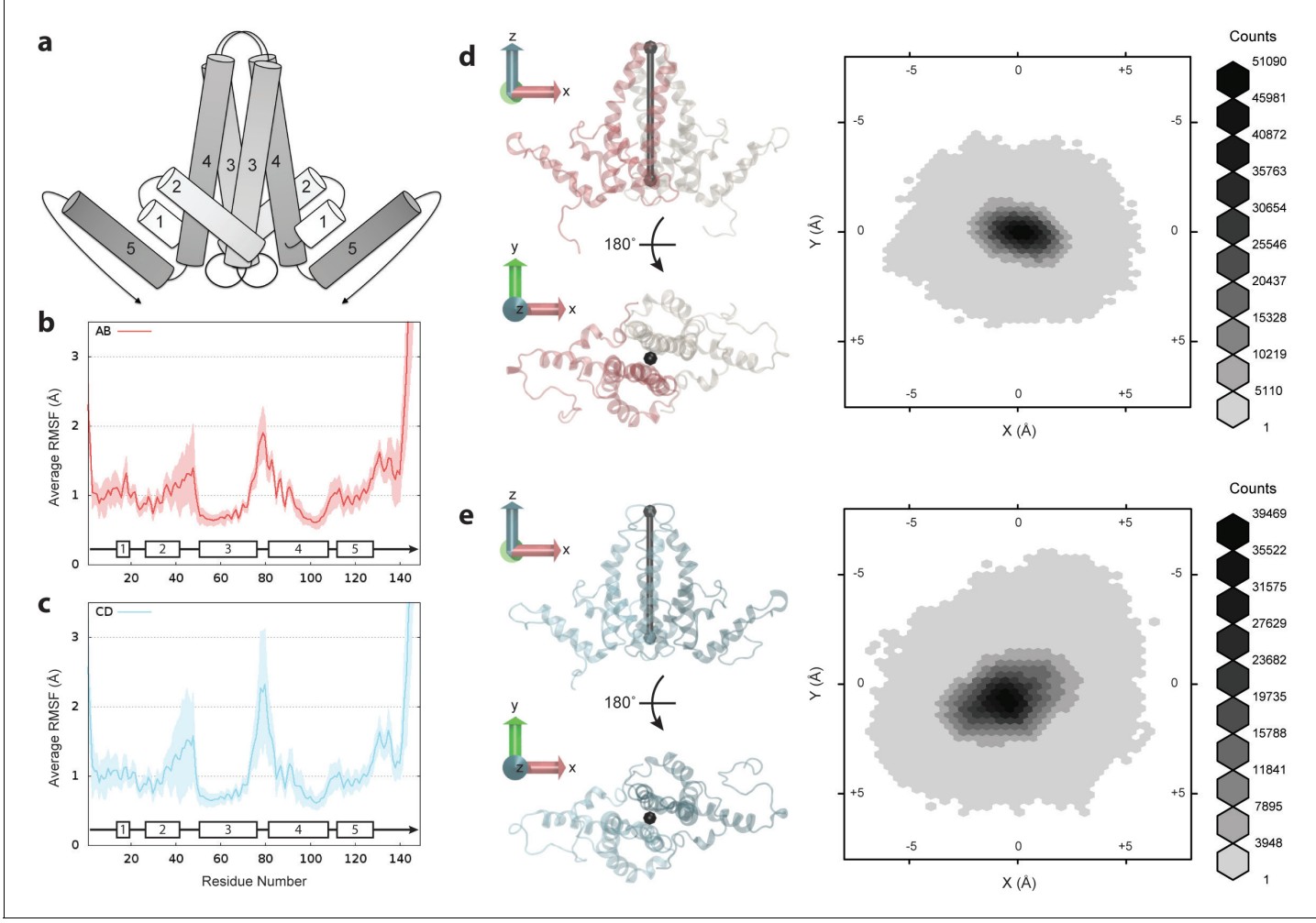

**Figure 2.** Flexibility of Cp dimers. (**a**) Schematic of Cp dimer, with helices 1 through 5 indicated. (**b**) Average Cα RMSF (Å) over 60 copies of AB dimers. (**c**) Average Cα RMSF (Å) over 60 copies of CD dimers. CD dimers are more flexible than AB dimers, with greater mobility in the spike tips. Calculations are based on internal alignment of dimers extracted from the capsid simulation, totaling 60 µs of conformational sampling for each dimer group. Error bars are given as standard deviation. (**d**) 3D histogram of AB dimer spike tip motions in the *xy* plane, given alignment of dimers along the *z*-axis. (**e**) 3D histogram of CD dimer spike tip motions in the *xy* plane, given alignment of dimers along the *z*-axis. CD dimer spike tips sample a larger spatial area than AB dimer spike tips. Histograms are based on 60 µs of conformational sampling for each dimer group, or a total of 3 M conformations. See Materials and methods for more information on dimer alignments and spike tip definitions.

DOI: https://doi.org/10.7554/eLife.32478.006

The following source data is available for figure 2:

**Source data 1.** Raw data for plots 2b-c in plain text format.
DOI: https://doi.org/10.7554/eLife.32478.007

residues 140 to 149 have been described as a linker (*Watts et al., 2002*) that connects the assembly domain to the disordered CTD, whose flexibility and mobility are known to be vital for capsid function throughout the infection cycle (*Venkatakrishnan and Zlotnick, 2016*).

Due to quasi-equivalence, dimers can be classified into two groups: those composed of A and B chains (AB dimers) and those composed of C and D chains (CD dimers). Comparing average Cα RMSF for dimer groups indicates that CD dimers are more flexible than AB dimers (*Figure 2b–c*). Apart from the C-termini, the spike tips at the junction of helices 3 and 4 and the region encompassing helix 2 and its connecting loop to helix 3 exhibit the greatest mobility. The flexibility and variability of the CD dimer exceeds that of the AB dimer in both of these key locations, demonstrating that local capsid dynamics are a function of quasi-equivalence.

Even within the crystal structure, CD dimers exhibit more disorder in the spike tips than AB dimers (*Bourne et al., 2006*). CD dimer spike tips also maintain a shorter gap between constituent chains than AB dimer spike tips (4.9 ± 4.4 Å versus 7.2 ± 5.2 Å from D78-Cα to D78-Cα' on average during simulation), indicating that greater mobility in this region does not arise from or lead to a more open dimer interface. An interesting observation from simulation is that spikes composed of both AB and CD dimers demonstrate the ability to sway back and forth. Measurement of fluctuations of spike tips in the *xy* plane based on alignment of dimers along the *z*-axis (described in Materials and methods) indicates that the motion of CD dimer spikes allows them to sample a larger spatial area than AB dimer spikes (*Figure 2d–e*). Because the structural features of viruses have been highly optimized to carry out specific functions necessary to drive infection, it is reasonable to hypothesize that the effects of quasi-equivalence on capsid dynamics could play an important, but as yet unknown role in capsid function.

## Global capsid dynamics are asymmetric

Beyond the internal motions of Cp dimers, the capsid exhibits global dynamics that arise from spatial movements of dimers relative to each other within the assembly. Comparison of average Cα RMSF measured for dimers extracted from the capsid (*Figure 3a*, bottom curve) with that of dimers considered within the context of the capsid (*Figure 3a*, top curve) clearly illustrates this observation.

The relative motion of dimers leads immediately to asymmetry within the capsid structure. In a symmetric capsid, dimer positions are related by standard transformation matrices, such as those available from the VIPER database (*Carrillo-Tripp et al., 2009*), which describe an idealized relationship between subunits. Capsid dynamics can cause deviations in the transformations between dimers of neighboring subunits in excess of ±8° (*Figure 3—figure supplement 1*), leading to spatial displacements of up to ±8 Å compared to a symmetric structure.

Projection of capsid RMSF values onto their corresponding locations in the capsid model illustrates a clear picture of asymmetry in global dynamics (*Figure 3b*). Given infinite conformational sampling (or icosahedral averaging), the projection might be expected to appear symmetric; however, combining the global motions sampled over the simulation timescale of 1 μs does not yield an idealized icosahedral structure. Beyond consistently high flexibility of the spikes, global dynamics appear non-uniform and follow no distinguishable pattern. RMSF values do not correlate with symmetry axes or correspond strongly to capsomere type.

Assessment of the local stability of capsomeres based on RMSD from the initial model indicates that hemaxers are slightly more flexible than pentamers on average (6.7 ± 0.3 Å versus 6.2 ± 0.2 Å), which can be attributed to the increased flexibility measured for CD dimers. This is in contrast to results for the HIV-1 capsid, whose hexamers were found to be notably more flexible than pentamers (3.8 ± 1.0 Å versus 2.7 ± 0.6 Å) (*Perilla and Schulten, 2017*).

Apparent asymmetry of capsomeres within the context of the capsid assembly arises from translations through space, which do not occur uniformly and are not fully sampled across symmetrically related regions of the capsid over 1 μs. Indeed, the time series for spatial fluctuations of capsid pentamers with respect to the capsid center reveals correlation, anti-correlation, and lack of correlation in the motions of symmetrically related pentamers (*Figure 4*). These observations indicate asymmetric distortions in capsid shape at equilibrium, and that capsid shape constantly fluctuates over long timescales.

## Essential dynamics emphasize asymmetric distortion

Due to the computational expense of simulating complete virus capsids, some alternative theoretical approaches have been utilized to predict dynamical and mechanical properties of capsids at reduced cost (*May, 2014*). In particular, normal mode analysis (NMA) has been applied to compute collective motions of icosahedral capsids, which have proven useful for suggesting pathways for large conformational transitions involving global expansion, such as swelling and maturation (*Tama and Brooks, 2002*; *Tama and Brooks, 2005*; *Kim et al., 2003*; *Rader et al., 2005*). To minimize computational expense, such calculations have relied on the use of symmetry constraints.

The availability of an unsymmetrized conformational ensemble produced by microsecond simulation of the HBV capsid provides the unique opportunity to examine capsid collective motions without the assumption of symmetry, as well as in the context of damping effects due to the presence of

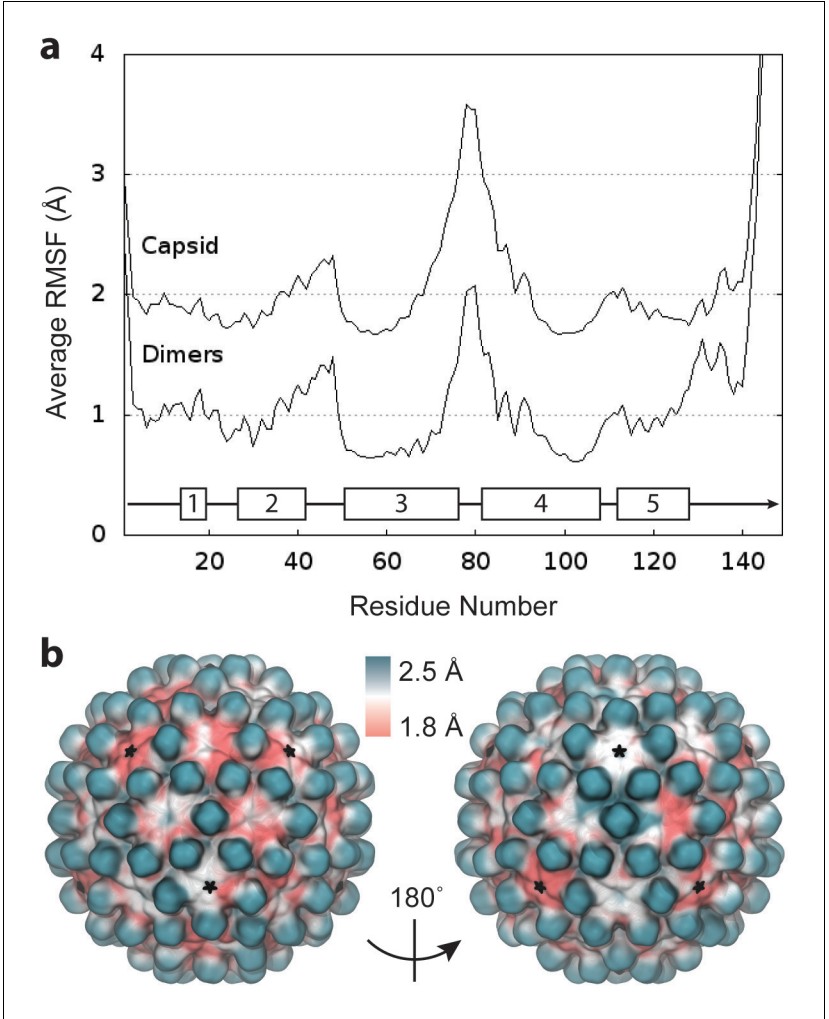

**Figure 3.** RMSF of complete capsid. (**a**) Average Cα RMSF (Å) of Cp dimers. Calculations are based on internal alignment of dimers extracted from the capsid simulation (local dynamics, bottom curve) and alignment of the full capsid (global dynamics, top curve), totaling 120 μs of conformational sampling. The increase in apparent dimer flexibility in the latter case arises from spatial movements of dimers relative to each other within the capsid assembly. (**b**) Projection of RMSF values fonto their corresponding locations in the capsid model instead of averaging over dimer copies reveals clear asymmetry in global dynamics. Fivefold vertices are highlighted with stars.
DOI: https://doi.org/10.7554/eLife.32478.008

The following source data and figure supplement are available for figure 3:

**Source data 1.** Raw data for plot 3a in plain text format.
DOI: https://doi.org/10.7554/eLife.32478.010

**Figure supplement 1.** Deviations in transformations between dimers of neighboring subunits.
DOI: https://doi.org/10.7554/eLife.32478.009

surrounding solvent. Essential dynamics of the capsid were determined by principal component analysis (PCA) of its Cα trace, using 50,000 conformations sampled over 1 μs of simulation. PCA results reveal remarkable complexity in the capsid's local and global dynamics and further emphasize asymmetric distortion in its overall shape. The lowest frequency mode (*Figure 5*) comprises only 6% of total variance; by the tenth mode, contribution to variance is less than 2% (*Figure 5—figure supplement 1*). More than 25 modes are required to account for 50% of the collective motions sampled by the capsid.

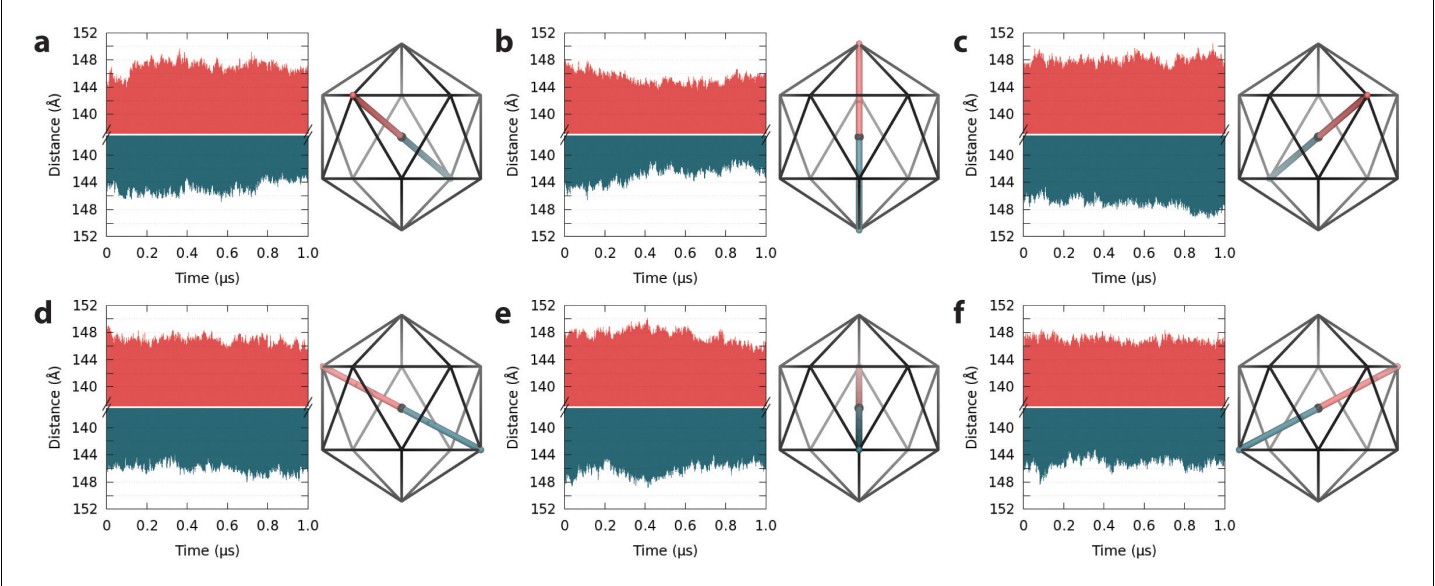

**Figure 4.** Spatial fluctuations of pentamers with respect to the capsid center. (a–f) Pentamers in the upper hemisphere (red) of the capsid are shown paired with their counterparts across the twofold symmetry axis in the lower hemisphere (blue). Icosahedral schematics indicate the relative locations of the measured distances (Å) within the capsid. Distances are measured between the center of mass of pentamers and the center of mass of the capsid. Variations in distance profiles indicate asymmetric distortions in capsid shape over time.

DOI: https://doi.org/10.7554/eLife.32478.011

The following source data is available for figure 4:

**Source data 1.** Raw data for plots 4a-f in plain text format.
DOI: https://doi.org/10.7554/eLife.32478.012

Previous work to predict collective motions of the HBV capsid based on NMA with symmetry constraints reported motions such as (1) an outward expansion of Cps on fivefold vertices coupled with inward expansion of Cps on threefold vertices, (2) capsid breathing, (3) twisting of Cps on fivefold and threefold vertices in opposite directions, and (4) pinching of Cps on fivefold and threefold vertices toward those vertices (*Dykeman and Sankey, 2010*). The modes computed in the present study include no assumption of symmetry and cannot be so simply described, as they encompass varied combinations of previously reported motions, a well as other movements, distributed non-uniformly across the capsid structure.

For example, in the lowest frequency mode (*Figure 5*), one region of the capsid exhibits anti-correlated inward and outward expansion of fivefold and threefold vertices, whereas another region exhibits similar motions in a correlated fashion. These motions are not necessarily radial in their expansion. In some cases, the motions stem from the Cps around sixfold vertices. While some capsomeres appear to move inward and outward, others seem to rock back and forth along the capsid surface; some expand and contract laterally; some twist; some undergo motion as complete capsomere units, while others shear or split. Capsid spikes exhibit some of the most dramatic movements, appearing to sway back and forth or move inward and outward radially. One common characteristic observed for all major modes was an elongation or compression of the capsid along one axis or another, resulting in the protrusion of some area of its surface and, thus, asymmetric distortion of its shape. An animation depicting the first three modes of capsid essential dynamics is provided in *Video 2*.

Notably, a symmetric breathing mode, which was previously predicted for the HBV capsid (*Dykeman and Sankey, 2010*), was not identified by PCA. Given infinite conformational sampling, the symmetric features of the capsid's essential dynamics might be expected to appear more pronounced. Future simulation work on icosahedral capsids that explore longer, multi-microsecond timescales, likely through the use of enhanced sampling methods, will shed further light on the role of symmetry versus asymmetry in capsid collective motions, particularly in the presence of solvent.

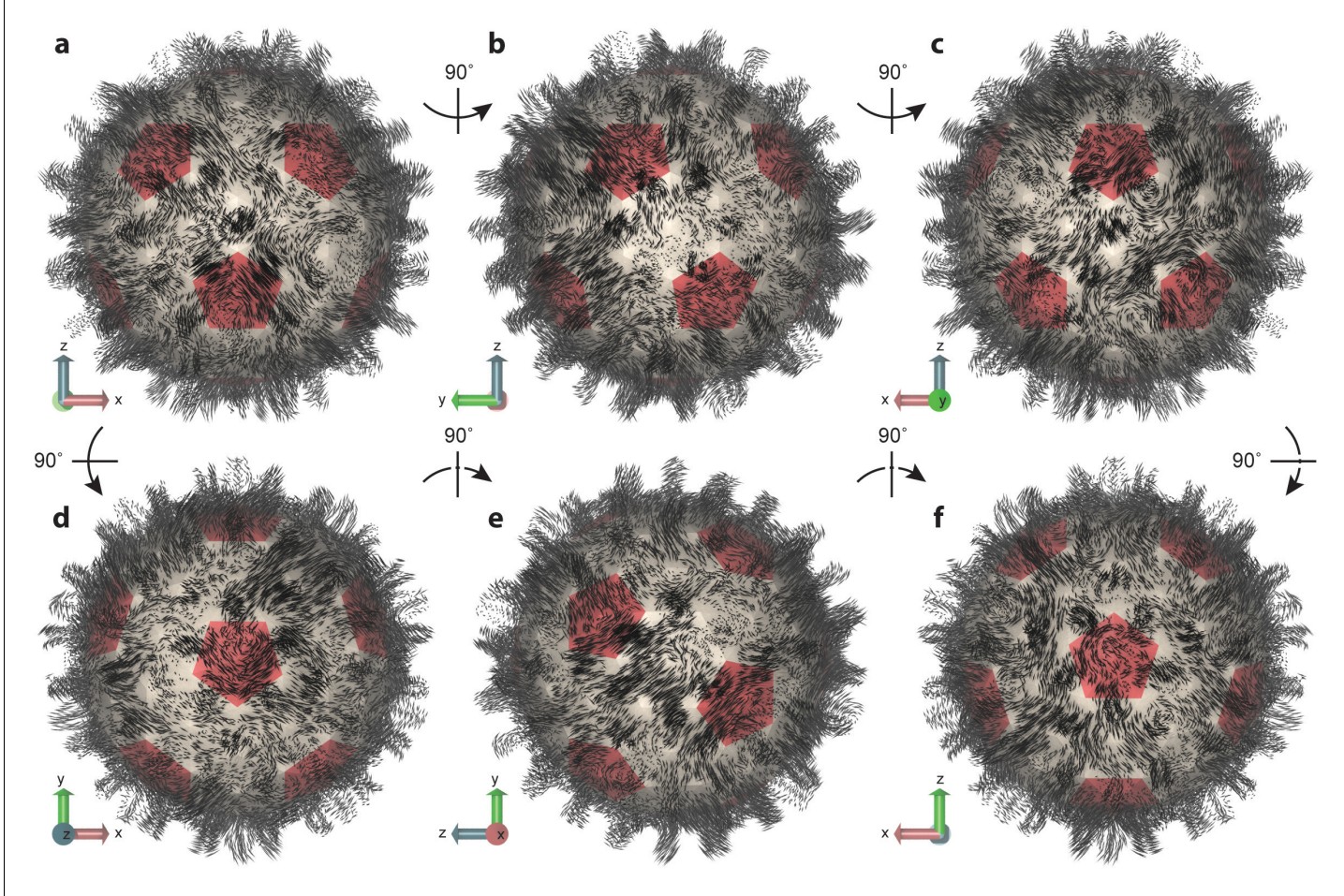

**Figure 5.** Essential dynamics of the capsid. (a–f) Views of the first mode from PCA, which comprises only 6% of total variance, illustrate the striking complexity and intrinsic asymmetry of capsid dynamics. PCA calculation based on Cα trace of 50,000 conformations sampled over 1 μs of simulation. Mode representation shown as a porcupine plot projected onto a polyhedral surface with pentamers highlighted in red; eigenvalue scaling increased by five for visual clarity.

DOI: https://doi.org/10.7554/eLife.32478.013

The following source data and figure supplement are available for figure 5:

**Source data 1.** Mode data in plain text format for NMWiz.

DOI: https://doi.org/10.7554/eLife.32478.015

**Figure supplement 1.** Scree plot for PCA modes.

DOI: https://doi.org/10.7554/eLife.32478.014

Importantly, some NMA studies of icosahedral capsids have suggested that the lowest frequency modes may actually be non-icosahedral, implying that asymmetric motions could be the most relevant to biological function (*Tama and Brooks, 2002*; *Rader et al., 2005*; *May et al., 2011*). However, more recent studies using multi-scale techniques that combine NMA with symmetry-constrained MD simulations suggest that, although asymmetric paths contribute to the ensemble of pathways available for expansion transitions, the symmetric paths dominate (*May et al., 2012*). Regardless of the role of symmetry in long timescale dynamics of the HBV capsid, the observation that the capsid is capable of asymmetric distortion, even under equilibrium conditions and in the absence of mechanical stress, raises the possibility that this may be an important biophysical feature, relevant to capsid function.

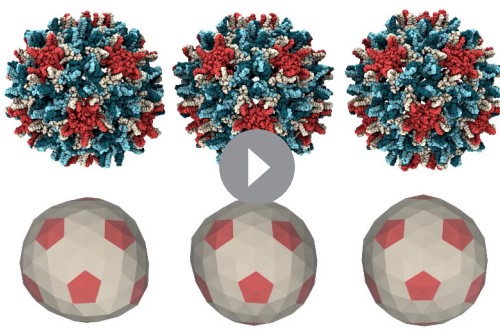

**Video 2.** Essential dynamics of the HBV capsid. Animation depicting the first three modes from PCA of the capsid, computed based on an ensemble of 50,000 conformations sampled over 1 µs of MD simulation. Modes are ordered from left to right and comprise 6.0%, 4.7%, and 3.4% contribution to total variance, respectively. Top panel illustrates the capsid's Cα trace, while the lower panel provides an abstraction based on a polyhedron used to describe changes in the capsid's global morphology. Animation rendered using NMWiz and VMD.
DOI: https://doi.org/10.7554/eLife.32478.016

# Capsid dynamics limit resolution in single-particle image reconstruction

Cryo-EM has gained new prominence as a method for determining the structures of large biomolecules, including the HBV capsid (*Conway et al., 1997*; *Yu et al., 2013*; *Wang et al., 2012*; *Schlicksup et al., 2018*), as sample preparation does not impose crystallographic packing constraints on molecular conformations, and single-particle image reconstructions now approach atomic resolution. Importantly, image reconstructions depend on averaging together many image samples of what is presumably a single biomolecular conformation; however, simulation results indicate that the HBV capsid, once thought of as a highly symmetric structure, actually explores a massive ensemble of asymmetric conformations under physiological conditions. The best cryo-EM resolution achieved thus far for the HBV capsid is 3.5 Å (*Yu et al., 2013*). Analysis of theoretical density maps calculated for conformations sampled during simulation demonstrate that the high flexibility of the capsid is a key factor limiting the resolution obtained for it in single-particle image reconstructions.

Density maps were calculated for 1,000 capsid conformations extracted at regular intervals over 1 µs of simulation time. The individual maps necessarily show atomic detail (*Figure 6a*). Aligning and averaging the 1,000 individual maps mimics the effect of performing a single-particle image reconstruction. The average map (*Figure 6b*) exhibits significant smoothing relative to the map calculated for a single conformation (*Figure 6a*), and the loss of detail clearly stems from the collective variability of capsid conformations included in the analysis. In addition to local flexibility of the capsid, such as the significant motions measured for spike tips, the distortions in capsid shape observed during simulation serve to reduce accuracy in structural alignment and blur atomic detail upon averaging. Differences in pentamer-pentamer diameter between individual maps were found to exceed 4 Å, which would be more than sufficient to cause smearing in experimental density. Applying icosahedral averaging, as is common for structure determination of virus capsids with both cryo-EM and crystallography, results in further degradation of atomic detail (*Figure 6c*). Given infinite conformational sampling, the maps from *Figure 6b and c* would likely exhibit yet less detail, but more closely resemble one another.

Resolution was estimated for the asymmetric image reconstruction represented in *Figure 6b* using Fourier shell correlation (FSC), which measures the correlation between two density maps as a function of spatial frequency. The data set used in the reconstruction (1,000 maps) was divided into two randomized subsets (500 maps). The maps in each subset were aligned and averaged, and the Fourier transforms of the new average maps were computed. The resolution of the reconstruction based on the full data set is defined where the FSC between the two sub-maps falls below a cutoff, typically 0.143 (*Rosenthal and Henderson, 2003*). The FSC between the two maps indicated a resolution of 2.3 Å (*Figure 6—figure supplement 1a*). For map calculation, the individual atoms were treated as Gaussian density distributions with a width (i.e. diameter) of 1 Å at half maximal density. As such, degradation of resolution to 2.3 Å represents a substantial loss of detail arising solely from the flexibility captured by 1,000 capsid conformations. The width specified in map calculation did not appear to introduce bias, as repeating the analysis with widths of 0.5 Å and 1.5 Å produced identical FSC curves (*Figure 6—figure supplement 1b*). Icosahedral averaging of the two sub-maps increased the correlation observed at lower spatial frequencies in the FSC curves, but did not alter

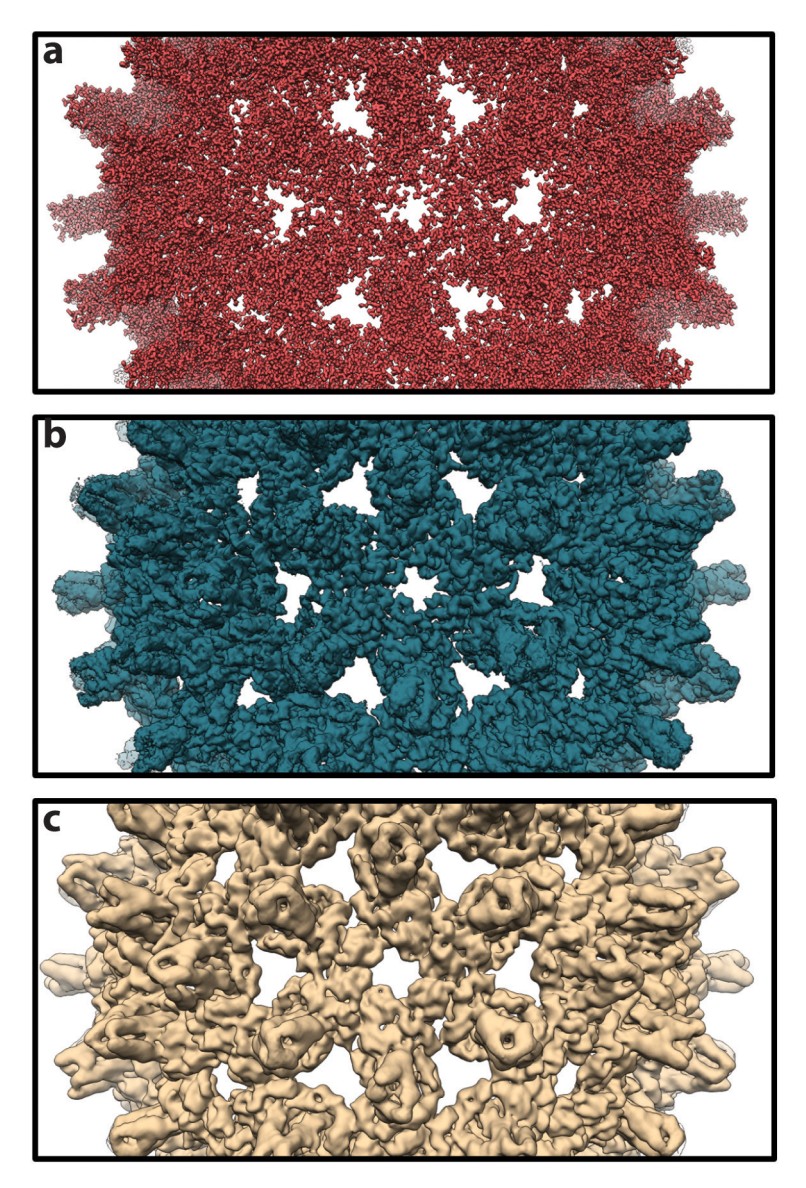

**Figure 6.** Theoretical density maps. (a) A representative density map calculated for a single conformer sampled during simulation. The map necessarily shows atomic detail, with individual side chains clearly visible. (b) An averaged map calculated as the mean of 1,000 individual maps, such as that in panel a, based on conformers extracted from the simulation at 1-ns intervals. All maps used in the calculation share a common orientation, based on alignment on the capsid's Cα trace. Averaging clearly reduces molecular detail. (c) A map calculated by icosahedral averaging of the map in panel b. With icosahedral averaging, a technique commonly used in experimental structure determination, molecular detail is further reduced and features at low spatial frequency become more apparent. For map calculations, individual atoms were treated as Gaussian density distributions with a width of 1 Å at half maximal density and pixel size of 0.75 Å.

DOI: https://doi.org/10.7554/eLife.32478.017

The following figure supplement is available for figure 6:

**Figure supplement 1.** FSC analysis of theoretical density maps.

DOI: https://doi.org/10.7554/eLife.32478.018

the predicted resolution of 2.3 Å (*Figure 6—figure supplement 1c*). While this resolution estimate is much better than the best cryo-EM resolution obtained thus far for the HBV capsid (3.5 Å [*Yu et al., 2013*]), the latter likely includes more extensive conformational sampling in the image reconstruction, as well as errors resulting from experimental considerations that are absent in the simulation study.

To further investigate the relationship between capsid dynamics and the results of single-particle image reconstruction, local resolution was estimated for the symmetric image reconstruction represented in *Figure 6c*. Local resolution provides information regarding structural order or relative atomic fluctuation within different regions of the capsid, analogous to a crystallographic B-factor (*Cardone et al., 2013*; *Wynne et al., 1999*). Similarly, RMSF characterizes variation in atomic coordinate positions arising from protein flexibility. For the HBV capsid, there is excellent correlation between RMSF and the calculated local resolution on a per-residue basis (*Figure 7a–c*). Notable correlation is also identified between RMSF and crystallographic B-factors compiled from the two

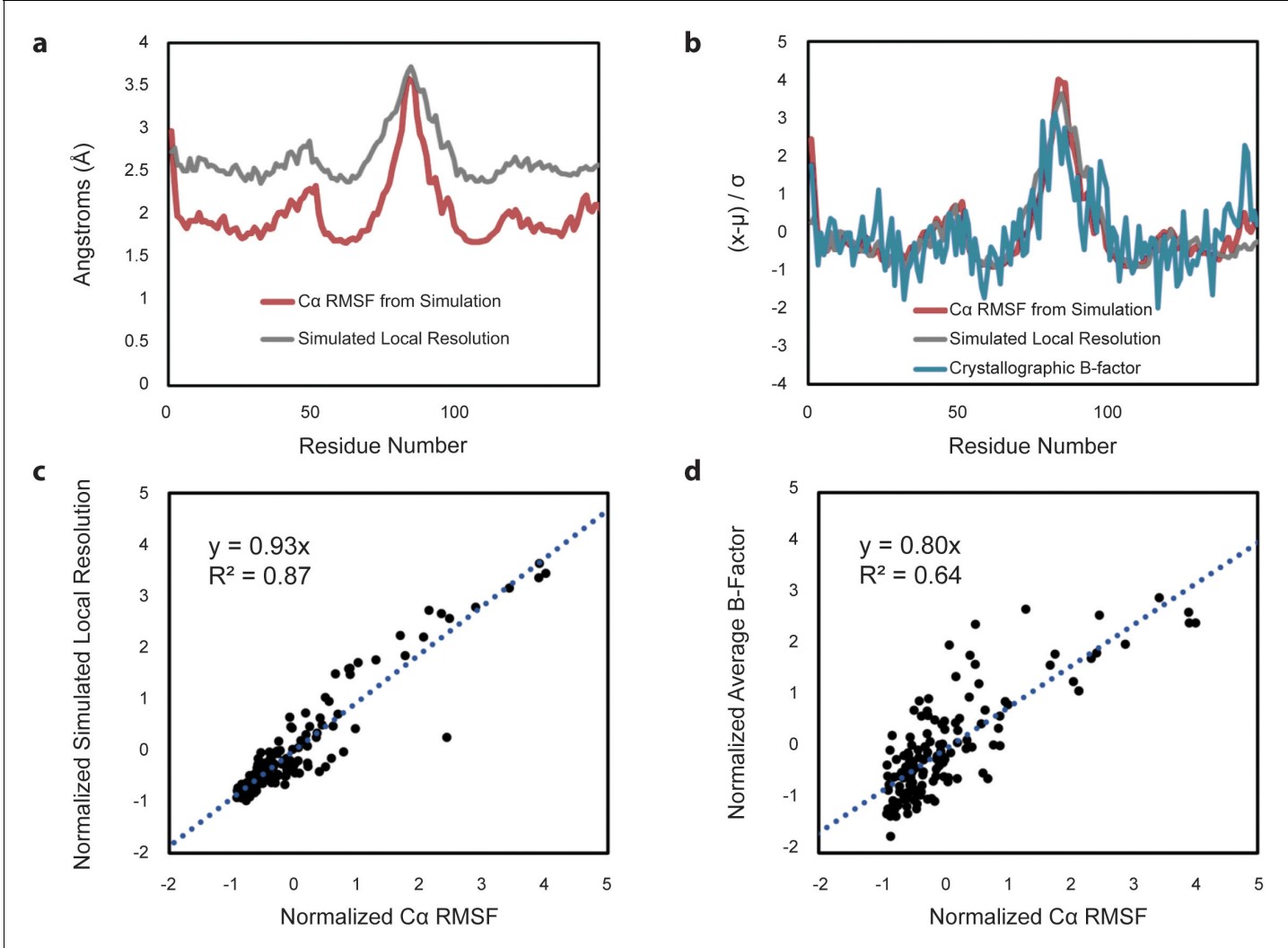

**Figure 7.** Comparison of Cα RMSF with theoretical local resolution and experimental B-factors. (a) Per-residue comparison of RMSF and local resolution; average RMSF is 2.03 Å, while average local resolution is 2.65 Å. For reference, the global FSC value of the theoretical density map is 2.3 Å. (b) Per-residue comparison of normalized RMSF and normalized local resolution with normalized experimental B-factors, reported as an average based on the two available apo-form HBV capsid crystal structures. To place metrics on the same scale for comparison, values (x) minus their average (μ) are normalized in units of standard deviation from the mean (σ). (c) Relative changes between normalized RMSF and normalized local resolution are highly correlated. (d) Relative changes between normalized RMSF and normalized experimental B-factors are also highly correlated, although noisier.
DOI: https://doi.org/10.7554/eLife.32478.019

available apo-form HBV capsid crystal structures (*Wynne et al., 1999*; *Bourne et al., 2006*) (*Figure 7b and d*). This result allows, for the first time, a direct line to be drawn between measurements of local dynamics made with three distinctly different structural techniques and confirms that the extent of local flexibility observed during simulation is supported by experiment.

## Capsid rapidly exchanges water molecules

Capsid dynamics occur within the context of the surrounding solution environment, primarily composed of water. Simulation results reveal that the HBV capsid is highly permeable to water, owing to its large pores. Measurement of exchange rates for solvent species crossing the capsid surface indicates that, under physiological conditions, water molecules exchange inward at a rate of $4.7 \times 10^3 \pm 0.03$ ns$^{-1}$ and outward at a rate of $4.7 \times 10^3 \pm 0.04$ ns$^{-1}$ (*Figure 8a*). These rates describe the transition from bulk solvent within the capsid to bulk solvent outside of the capsid, and vice versa, defined by a distance of $\pm 20$ Å from the capsid's average radius. Equivalence between inward and outward rates confirms that solvent exchange is in equilibrium between the capsid interior and exterior. The rate of exchange of water across a spherical surface with the same average radius as the capsid is estimated to be $233.1 \times 10^3 \pm 2.1$ ns$^{-1}$ (*Figure 8—figure supplement 1a*). All pore types appear to be capable of bidirectional water transport, with, larger pores responsible for exchanging more molecules of water per unit time (triangular pores > hexameric pores > pentameric pores).

The measured exchange rates for water contrast starkly with results from simulation studies of other capsids. The poliovirus capsid (T = 4, 30 nm) was found to exchange water molecules at an average rate of only $8 \pm 2$ ns$^{-1}$ (*Andoh et al., 2014*), indicating relatively low permeability compared to the HBV capsid. The smaller porcine circovirus type 2 (PCV2) capsid (T = 1, 20 nm) was found to exchange water molecules at an average rate of $73 \pm 3$ ns$^{-1}$ (*Tarasova et al., 2017b*), while the satellite tobacco necrosis virus (STNV) capsid (T = 1, 22 nm) exhibited a rate of 10 ns$^{-1}$ (*Larsson et al., 2012*). The significantly larger HIV-1 capsid exchanged water molecules at an average rate of $19.8 \times 10^3 \pm 1.6$ ns$^{-1}$ (*Perilla and Schulten, 2017*). Interestingly, the rates measured for these simulated capsids span three orders of magnitude, yet do not correlate with the trend of capsid size or surface area. Such an observation highlights the remarkable diversity of capsid function across unrelated viruses.

## Capsid exchanges sodium five times faster than chloride through triangular pores

Under physiological conditions, the capsid's solvent environment also contains charged ionic species, particularly 150 mM NaCl. Simulation results reveal that, consistent with findings for other viruses, the HBV capsid serves as a semipermeable membrane structure that selectively filters water and ions. Measurement of exchange rates for ionic species crossing the capsid surface indicates that, at 150 mM NaCl, sodium ions exchange inward and outward at rates of $8.4 \pm 0.7$ ns$^{-1}$ and $8.5 \pm 0.6$ ns$^{-1}$, respectively (*Figure 8b*), and chloride ions exchange inward and outward at rates of $1.8 \pm 0.4$ ns$^{-1}$ and $1.7 \pm 0.3$ ns$^{-1}$, respectively (*Figure 8c*). Equivalence between inward and outward rates confirms that ion exchange is, like water exchange, in equilibrium between the capsid interior and exterior. The rate of exchange of ions across a spherical surface with the same average radius as the capsid is estimated to be $412.3 \pm 93.3$ ns$^{-1}$ for sodium (*Figure 8—figure supplement 1b*) and $516.6 \pm 121.1$ ns$^{-1}$ for chloride (*Figure 8—figure supplement 1c*). An animation depicting exchange of ions from the capsid interior to exterior is provided in *Video 3*.

Remarkably, the rate of exchange of sodium across the capsid surface is nearly five times faster than that of chloride. Due to the highly negative net charge of the Cp149 capsid ($-1680e$ at pH 7.0), an excess of sodium ions was included in the simulation to achieve neutrality, such that there is a sodium to chloride ratio of approximately 4:3. While the presence of more sodium in the system could account for the observation of more translocation events for this species, a preference to exchange sodium rather than chloride remains clear.

The measured rate for sodium exchange is consistent with the average rate determined for the HIV-1 capsid ($9.0 \pm 1.9$ ns$^{-1}$), yet HIV-1 was found to preferentially exchange chloride (*Perilla and Schulten, 2017*). Exchange of chloride ions across the HIV-1 capsid surface proceeded at an average rate more than twice that of sodium ($21.5 \pm 3.1$ ns$^{-1}$) and was shown to occur through charge-specific channels at the centers of capsomeres (*Perilla and Schulten, 2017*). 3D potentials of mean force

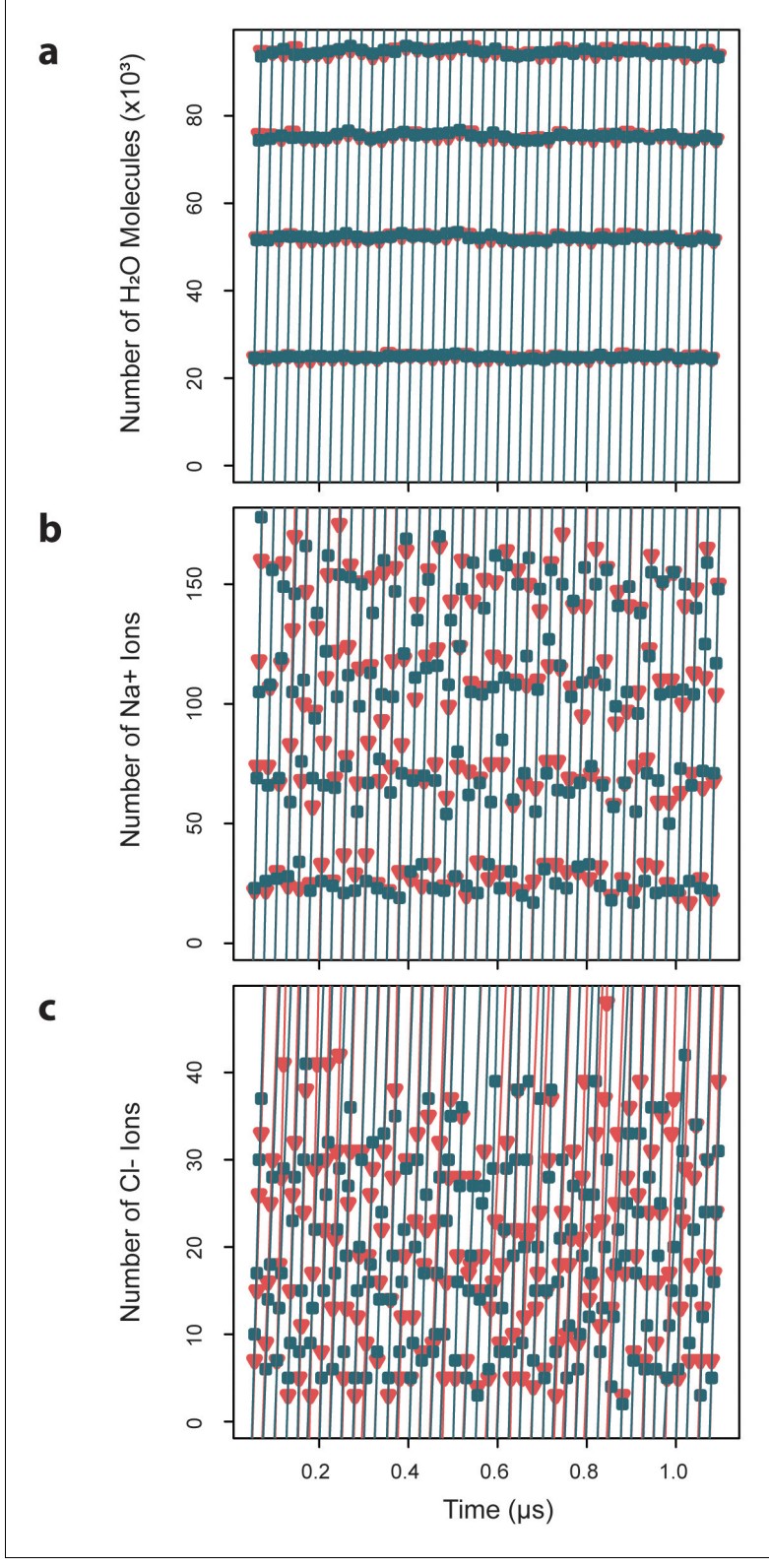

**Figure 8.** Exchange rates for water and ions crossing the capsid surface. Cumulative numbers of solvent species moving inward (blue) and outward (red) across the capsid surface over a given segment of simulation time are plotted versus that simulation time, and the slope of the linear fits give the exchange rates, which are reported as averages ± standard deviations. (a) Water molecules exchange at a rate of $4.7 \times 10^3 \pm 0.03$ ns$^{-1}$ inward and

*Figure 8 continued on next page*

*Figure 8 continued*

$4.7 \times 10^3 \pm 0.04$ ns$^{-1}$ outward. (**b**) Sodium ions exchange at a rate of $8.4 \pm 0.7$ ns$^{-1}$ inward and $8.5 \pm 0.6$ ns$^{-1}$ outward. (**c**) Chloride ions exchange at a rate of $1.8 \pm 0.4$ ns$^{-1}$ inward and $1.7 \pm 0.3$ ns$^{-1}$ outward.

DOI: https://doi.org/10.7554/eLife.32478.020

The following figure supplement is available for figure 8:

**Figure supplement 1.** Exchange rates for water and ions crossing a spherical surface.
DOI: https://doi.org/10.7554/eLife.32478.021

derived from ion occupancy maps indicated that chloride translocates more easily through hexameric channels (*Perilla and Schulten, 2017*), which far outnumber pentameric channels in the HIV-1 capsid, partially accounting for an enhanced rate of chloride exchange. In contrast to both HBV and HIV-1, simulation studies of the poliovirus and PCV2 capsids suggested that, while both structures are permeable to water, they do not exchange ions (*Andoh et al., 2014*; *Tarasova et al., 2017b*).

Ion translocation events were observed for all pore types in the HBV capsid during simulation, including those at the centers of capsomeres; however, computed ion occupancy maps reveal strong localization of sodium only within the triangular pores that occur on either side of CD dimers, indicating that sodium exchange primarily occurs through these apertures. Occupancy values are directly related to the Gibbs free energy of binding ($\Delta G$) (*Cohen et al., 2006*), and a 3D potential of mean force was derived to quantify the strength of ion association based on the ratio of species occupancy at a given site versus its occupancy in bulk solvent. At $\Delta G = -1.0$ kcal mol$^{-1}$, sodium localization is prominent within the triangular pores, where three copies each of D2, D4, E14, D40, and E43 aggregate from the surrounding trimers of dimers (*Figure 9a*). The highly negative nature of the triangular pores likely relates to their preference to exchange positive ions and may contribute to the relatively slow rate of exchange observed for negative ions. Paradoxically, permeability of the HBV capsid to negatively charged species is essential for maturation, when nucleotides must pass in and out through capsid pores to support reverse transcription.

## Sodium shell forms along capsid interior

HBV capsid assembly is known to be responsive to solution conditions (*Kukreja et al., 2014*), and our results indicate that the capsid can in turn induce changes in the properties of its surrounding environment. Notably, computed ion occupancy maps reveal that the capsid invites the localization of sodium ions in key regions around its structure. At $\Delta G = -0.5$ kcal mol$^{-1}$, sodium localizes in a shell along the capsid's interior surface and in arcs across the capsid spike tips where two copies each of E77 and D78 come together in close proximity (*Figure 9b*). Occupancy is absent along the interior face of dimers themselves, yet most prominent along the interior face of dimer-dimer contacts. The sodium shell is apparent up to $\Delta G = -0.8$ kcal mol$^{-1}$, at which point it more clearly resembles a lattice.

The observation of a weakly associated sodium shell along the interior surface of the HBV capsid contrasts starkly with results from simulation studies of other capsids. Chloride was instead noted along the interior of both the HIV-1 (*Perilla and Schulten, 2017*) and PCV2 capsids, with a sufficient 'neutralizing' layer of

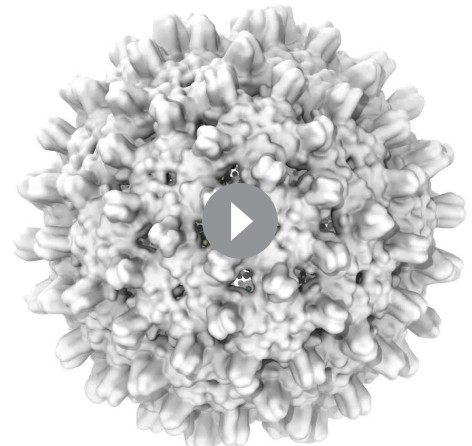

**Video 3.** Exchange of NaCl across the HBV capsid. Animation depicting the motions of ions contained within the capsid at the beginning of MD simulation over 0.1 µs of system equilibration. Ions exchange from the capsid interior to exterior over time. Due to the capsid's preference to exchange positive ions, more sodium (yellow) exits the capsid than chloride (cyan). Animation rendered using VMD.
DOI: https://doi.org/10.7554/eLife.32478.022

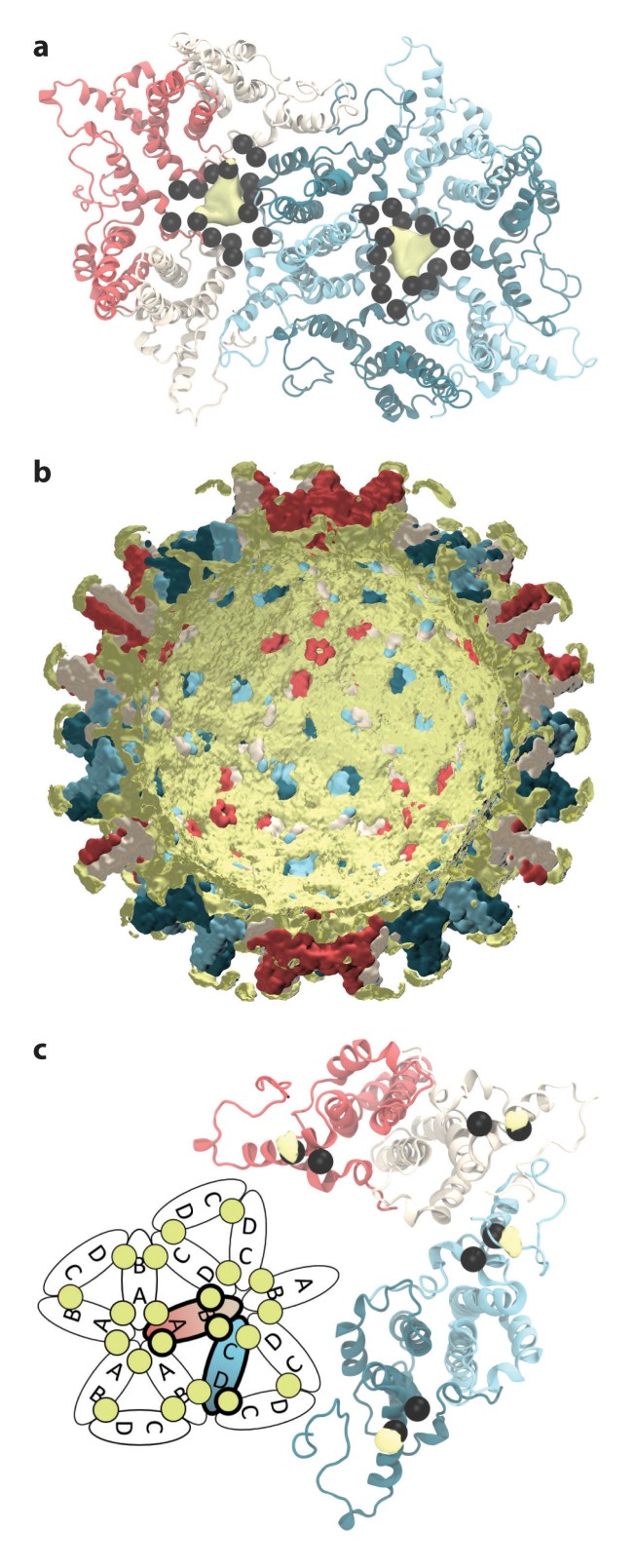

**Figure 9.** Sodium ion occupancy. (a) View of two AB and three CD dimers from the capsid exterior, illustrating sodium ion localization (yellow) within the pores at the center of trimers of dimers, where three copies each of D2, D4, E14, D40, and E43 (black spheres) aggregate. Isosurface contour level at ΔG = −1.0 kcal mol$^{-1}$. Calculation based on alignment of 60 asymmetric units with surrounding solvent, totaling 60 µs of conformational sampling.
*Figure 9 continued on next page*

*Figure 9 continued*

(b) Cross-section of the capsid showing sodium ion localization (yellow) in a shell along the interior and in arcs above the spike tips. Isosurface contour level at $\Delta G = -0.5$ kcal mol$^{-1}$. Calculation based on alignment of full capsid with surrounding solvent, totaling 1 μs of conformational sampling. (c) View of the asymmetric unit from the capsid interior, illustrating sodium binding (yellow) at E117, adjacent to E113 (black spheres). Isosurface contour level at $\Delta G = -1.4$ kcal mol$^{-1}$. Calculation based on alignment of 60 asymmetric units with surrounding solvent, totaling 60 μs of conformational sampling. Inset: Schematic showing sodium binding locations within extended capsid lattice. $\Delta G$ error estimates are all within $4\times10^{-5}$ kcal mol$^{-1}$.

DOI: https://doi.org/10.7554/eLife.32478.023

The following figure supplement is available for figure 9:

**Figure supplement 1.** Chloride ion occupancy.

DOI: https://doi.org/10.7554/eLife.32478.024

chloride shown to be critical for the stability of PCV2 (*Tarasova et al., 2017a*). Sodium was found to prefer the exterior surface of the HIV-1 capsid (*Perilla and Schulten, 2017*).

The highly negative net charge of the Cp149 capsid ($-1680e$ at pH 7.0) could account partially for the observed affinity of sodium ions for the capsid surface. In the Cp183 capsid, the disordered CTD tails carry a highly positive net charge (+3600e at pH 7.0) and have been shown to localize primarily along the interior surface of the capsid (*Selzer et al., 2015*), roughly corresponding to the shell region weakly occupied by sodium ions during simulation. The ability of the capsid to closely corral positive species along its interior likely contributes to the organization of the CTD tails within the capsid, explaining their experimentally observed preference to remain there and become exposed to the capsid exterior only transiently for cellular signaling (*Selzer et al., 2015*).

In a mature Cp183 capsid containing a 3200-basepair dsDNA genome, the negative charge of phosphates ($-6400e$ at pH 7.0) should massively exceed the positive charge contributed by CTD tails. Experimental work on Cp183 phosphorylation-mimic mutants indicate that the presence of phosphates increases the preference of the CTD tails to remain within the capsid (*Selzer et al., 2015*). Combined with experimental observation of a notable gap between the genome and capsid surface (*Wang et al., 2014*), these findings suggest that the putative sodium shell identified by simulation is a feature of the native capsid structure and may serve to electrostatically stabilize phosphorylated CTD tails within the capsid interior. The CTD tails likely displace the sodium shell in the absence of phosphorylation.

## Sodium binds to Cp dimers

Further inspection of occupancy maps indicates that sodium ions can bind directly to Cp dimers. At $\Delta G = -1.4$ kcal mol$^{-1}$, specific sodium binding is apparent at E117, where helix 5 extends from the dimer-dimer interface (*Figure 9c*). E113 is adjacent on the helix, and simulation results demonstrate that the localization of sodium here can transiently bridge interactions with E145 from the C-termini of both the same and contacting dimer within the capsomere. Interactions between helix 5 and E145 were not previously emphasized by experiment, as the flexible C-termini of Cp149 are not resolved in crystal structures. Specific sodium binding was also identified in the HIV-1 capsid, within the three-fold interfaces between hexamers, and shown to bind the capsid structure with an affinity similar to that observed for HBV ($\Delta G = -1.8$ kcal mol$^{-1}$) (*Perilla and Schulten, 2017*).

## Chloride mediates intra-dimer interactions

While simulation results suggest that sodium is highly involved in HBV capsid function, chloride ions were observed to play but a single role based on occupancy maps, weakly mediating an intra-dimer contact. At $\Delta G = -0.8$ kcal mol$^{-1}$, chloride localization is associated with R28 and R39 on helix 2 from one Cp chain and the positively charged N-terminal M1 from its partner chain within the dimer (*Figure 9—figure supplement 1*). The contact to helix 2 bridged by chloride likely serves to reduce mobility of the flexible N-terminal region that precedes helix 1, thus enhancing structural stability of the capsid. Specific chloride binding was identified in the HIV-1 capsid, within the central pores of capsomeres, but was shown to bind the capsid structure with a higher affinity than observed for HBV ($\Delta G = -1.5$ kcal mol$^{-1}$ in hexamers and $\Delta G = -2.7$ kcal mol$^{-1}$ in pentamers) (*Perilla and Schulten, 2017*).

## Discussion

Simulation of the HBV capsid on the microsecond timescale reveals a highly flexible structure, whose global dynamics are intrinsically asymmetric. Analysis of HBV capsid shape and essential dynamics identifies a common theme of subtle ellipsoidal distortion, which occurs even at equilibrium. Cryo-EM and biochemical studies indicate that mature HBV capsids (36 nm in diameter) can pass through the nuclear pore (39 nm in diameter) to become localized within the nuclear basket, even when associated with importin α/β (*Panté and Kann, 2002*; *Gallucci and Kann, 2017*). It may be that plasticity in overall capsid structure is necessary to facilitate its ability to maneuver through the relatively close-fit of the nuclear pore's central channel. Indeed, coarse-grained MD simulations, combined with atomic force microscopy experiments, showed that mechanical deformation of the capsid was reversible up to 60% of its average radius (*Arkhipov et al., 2009*), demonstrating the capsid's ability to undergo moderate compression without structural failure. The capsid's high permeability to water, owing to its large pores, may confer the ability to rapidly equilibrate water density across its surface upon mechanical deformation to prevent bursting.

Further, the maturation of HBV cores, which involves reverse transcription of pgRNA to dsDNA within the assembled capsid, has been associated with structural destabilization. This effect is believed to be due, at least in part, to mechanical strain imposed by the increased rigidity and organization of dsDNA (*Cui et al., 2013*; *Dhason et al., 2012*; *Guo et al., 2010*). Simulation results indicating that the capsid is highly flexible and capable of asymmetric distortion support speculations that it may flex or distort asymmetrically to accommodate the uneven distribution of internal pressure arising from genome maturation, in turn rendering the capsid metastable (i.e. spring-loaded) to facilitate uncoating (*Dhason et al., 2012*). The ability of the capsid to filter negatively charged ionic species and translocate them at a relatively slow rate could serve to regulate the rate of reverse transcription, such that structural distortion to accommodate dsDNA is allowed to occur gradually.

A growing body of experimental evidence indicates that the capsid's mechanical properties are essential to its biological function. Mutations in the Cp assembly domain, distant from the packaged pgRNA, result in a capsid that supports reverse transcription of only one strand of the viral genome, suggesting that capsid dynamics play an important role in genome replication (*Tan et al., 2015*). A novel treatment strategy for HBV may, thus, lie in the development of small-molecule drugs designed to increase capsid rigidity or otherwise limit the relative degrees of freedom of subunits, such that key dynamical properties are suppressed or any necessary asymmetric distortion becomes impossible. Recent work has shown that a class of HBV capsid-directed antiviral compounds not only modulate assembly, but also cause pre-formed capsids to become asymmetric (*Perilla et al., 2016*; *Schlicksup et al., 2018*), underscoring the ability of drugs to alter capsid structure and further demonstrating the capsid's readiness to undergo distortion in response to external stress.

Mechanical malleability of the HBV capsid has been attributed to relatively loose packing of its constituent Cp dimers compared to packing of other capsids (*Arkhipov et al., 2009*). The complex dynamics observed during simulation, as well as variability in collective motions across the capsid within single modes, may result from such loose subunit packing, relatively weak interaction energies between capsid subunits (*Ceres and Zlotnick, 2002*), and/or a relatively flat energy surface for different inter-subunit interactions, all of which can contribute to global flexibility and, thus, asymmetry. Moreover, structural diversity arising from capsid flexibility and asymmetry is likely critical to viral function. Conformational variations contribute to entropic stabilization of the capsid, which may be an important component of the hysteresis between assembly and disassembly reactions. Entropic mechanisms are now known to play an important role in allosteric regulation and protein-protein interaction (*Hilser et al., 2012*). Such transitions will further contribute to low-frequency structural events, including reverse transcription and conformational changes that the virus uses to surveil its environment, such as exposure of nuclear localization signals.

From a cryo-EM perspective, asymmetry in virus particles can be defined in terms of a distinct structural feature (such as a bound receptor), structural defects, or stochastic dynamical effects (*Wang et al., 2018*). There have been a number of asymmetric reconstructions of icosahedral viruses that treat asymmetry with respect to a structural feature, including HBV containing pgRNA and reverse transcriptase (*Wang et al., 2014*), canine parvovirus bound to a single receptor (*Hafenstein et al., 2007*), MS2 bound to a pilus (*Dent et al., 2013*), and MS2 with genome (*Dai et al., 2017*). Asymmetry with respect to particle defects was explored in a recent analysis of

Ross River virus (*Wang et al., 2015*). The present work has enabled investigation of the effects of asymmetry driven by protein dynamics.

Density maps calculated for conformations sampled during simulation were averaged to yield a final map that incorporates structural heterogeneity and mimics the result of a single-particle image reconstruction. Despite the all-atom level of detail encompassed by simulation, capsid flexibility and asymmetry were sufficient to cause notable blurring of the final map, particularly when icosahedral averaging was applied. Resolution of the theoretical image reconstruction was 2.3 Å, while the best cryo-EM resolution achieved thus far for the HBV capsid is 3.5 Å (*Yu et al., 2013*). Importantly, the latter value contains errors resulting from experimental considerations that lower resolution, such as heterogeneity due to sample preparation, the inherently low signal-to-noise ratio of imaged particles, error in particle alignment, inelastic scattering, incomplete sampling of particle orientations, and imperfections in the experimental hardware. Yet, in the absence of experimental error, simulation results indicate that thermal motion of particles makes large contributions to limiting the resolution of image reconstructions.

Single-particle image reconstructions depend on averaging together many image samples of what is presumably a single biomolecular conformation; however, simulation results indicate that icosahedral capsids can explore a massive ensemble of asymmetric conformations. Averaging over such an ensemble, even without imposing icosahedral symmetry, will force the result to appear as a single, blurry state. New 3D classification techniques can separate substantially different states, but the structural variation and diversity observed during simulation suggests a continuum of states, which will be difficult for image processing software to distinguish. Capsid dynamics may also lead to a large fraction of images being discarded due to particle heterogeneity. Thus, even in this era of increasing cryo-EM resolution, intrinsic capsid flexibility may represent a major limiting factor to achieving true atomic (1–2 Å) resolution. Symmetry relaxation methods may warrant greater consideration in future work.

Capsid dynamics occur within the context of the surrounding solution environment. Simulation results indicate that the capsid can induce changes in the properties of its environment, particularly with respect to the behavior of ions. Analysis of solvent exchange rates reveal that the capsid selectively filters ions based on charge and translocates sodium across its surface more rapidly than chloride. The ability of the capsid to differentiate ionic species and preferentially translocate those that are positively charged may play a role in facilitating the extrusion of the positively charged CTD tails through the capsid's pores to expose them to the surface for cellular signaling. By the same logic, the addition of negative charges on the CTD tails by phosphorylation could serve to slow the rate of extrusion, accounting in part for the observation that Cp183 phosphorylation-mimic mutants show reduced rates of proteolytic cleavage, consistent with reduced exposure to the capsid exterior (*Selzer et al., 2015*). Thus, based on its ion exchange preferences, the capsid could employ phosphorylation and dephosphorylation to modulate its own cellular signaling process.

Ion occupancy maps indicate that the translocation of sodium ions primarily occurs through the triangular pores located on either side of CD dimers, suggesting that these apertures may also represent the gateways of CTD tail extrusion. Cryo-EM studies have implicated the capsid hexamers as the sites of extrusion based on structural observation of a protein kinase within the hexameric valleys of the capsid exterior, presumably bound there to CTD tails (*Chen et al., 2011*). These separate findings are not mutually exclusive. Hexameric valleys are each surrounded by six triangular pores, and as the largest valleys between capsid spikes, they may represent the most accessible location along the surface for close approach of a kinase for CTD tail binding. Further, the triangular pores are the largest surface apertures, and are, thus, best able to accommodate translocation of the peptide chains that comprise CTD tails. The swaying motion of capsid spikes identified by simulation could function to guide CTD tails extruded through triangular pores to their neighboring hexameric valleys for interaction with kinases for phosphorylation. Enhanced flexibility of CD dimer spikes relative to AB dimer spikes could bias the movement of CTD tails to hexameric valleys rather than pentameric valleys, which may be too spatially restrictive to accommodate kinase approach.

If the capsid's triangular pores are indeed the location of CTD extrusion and therefore play a critical role in cellular signaling, a novel treatment strategy for HBV may lie in the development of a peptide designed to block the pore to prevent extrusion and inhibit signaling. Recent work on the HIV-1 capsid demonstrated the tractability of pores as drug targets, describing a channel inhibitor that blocks the pores to prevent nucleotide translocation into the capsid and, thereby, suppresses

reverse transcription (*Jacques et al., 2016*). Pore-blocking drugs could also serve to inhibit reverse transcription in the HBV capsid.

Strong localization of sodium ions observed within the capsid's triangular pores appears to enhance dimer-dimer interaction in regions of high negative charge. Positively charged metal ions have been shown experimentally to increase the rate of capsid assembly (*Ceres and Zlotnick, 2002*; *Stray et al., 2004*; *Choi et al., 2005*), possibly through specific binding to Cp (*Ceres and Zlotnick, 2002*; *Stray et al., 2004*). Zinc and nickel have been shown to accelerate assembly at much lower concentrations than monovalent cations and are hypothesized to function by binding Cp dimers and inducing them to adopt an assembly active conformation; a putative binding site for zinc ions has been proposed (*Stray et al., 2004*). However, simulation does not identify sodium localization within the zinc binding site. Importantly, zinc and nickel are transition metals capable of forming coordination spheres, while sodium is an alkali metal whose ionic interactions are governed primarily by electrostatics. As kinetic enhancement by divalent transition metals is conferred at lower concentrations than other divalent ions that lack coordination ability, such as calcium and magnesium (*Stray et al., 2004*), it is likely that the mechanism by which zinc and nickel accelerate assembly acts in addition to that of sodium or other non-transition metals.

Simulation results suggest that sodium reduces electrostatic repulsion within the triangular pores where three dimers pack together to form a trimer of dimers. Importantly, a trimer of dimers is implicated as the nucleation core for HBV capsid assembly (*Zlotnick et al., 1999*), and electrostatic stabilization of this critical substructure may represent a mechanism by which positively charged metal ions enhance assembly kinetics. Such an effect is consistent with calculations suggesting that ionic strength promotes HBV capsid assembly by globally screening electrostatic repulsion between dimers (*Kegel and Schoot Pv, 2004*). Interestingly, sodium binding was also observed within a trimeric dimer interface in the HIV-1 capsid (*Perilla and Schulten, 2017*), whose assembly is likewise thought to be nucleated by a trimer of dimers (*Grime and Voth, 2012*; *Tsiang et al., 2012*).

## Materials and methods

### Computational modeling

An all-atom model of the HBV capsid assembly domain was constructed based on an available apo-form crystal structure (PDB ID 2G33) (*Bourne et al., 2006*). Unresolved residues at the C-terminus of each chain were modeled using ROSETTA (*Leaver-Fay et al., 2011*) to produce a complete Cp149 capsid. The termini of five A chains within a capsid pentamer were folded simultaneously to generate 2,000 structures. The termini of two each of B, C, and D chains within a capsid hexamer were folded simultaneously to generate 5,000 structures. Two residues preceding the modeled termini were allowed flexibility in each case, while all other crystallographic residues were held fixed. Altogether, this procedure produced an ensemble of 10,000 structural models for each chain terminus within the asymmetric unit. The ensembles were clustered based on protein backbone RMSD, employing a partitioning around medoids scheme, as implemented by (*Goh et al., 2015*). The medoid for the most populated cluster from each ensemble was taken for the final model for A, B, C, and D chain termini, respectively.

### System preparation

Hydrogen positions and protonation states were assigned to the capsid for pH 7.0 using the propKa (*Søndergaard et al., 2011*) option of PDB2PQR 2.0.0 (*Dolinsky et al., 2007*). The CIonize plugin of VMD 1.9.2 (*Humphrey et al., 1996*) was used to place sodium and chloride ions around the capsid structure according to its local electrostatic potential. The capsid's net charge is $-1680e$, and a total of 1680 local sodium and 20 local chloride ions were placed. The capsid, along with local ions, was then suspended in a 392 $Å^3$ box of explicit solvent with bulk ion concentration of 150 mM NaCl to produce a charge-neutral system of nearly 6 M atoms. Box size was selected to ensure a distance of at least 30 Å was maintained between periodic copies of the capsid during simulation. The capsid system was parameterized with the CHARMM36 (*MacKerell et al., 1998*; *MacKerell et al., 2004*) force field and TIP3P (*Jorgensen et al., 1983*) water model. Simulation files were prepared using the psfgen plugin in VMD.

## Molecular dynamics simulations

Simulations were performed with NAMD 2.11 (*Phillips et al., 2005*) on the NSF Blue Waters super-computer. A steepest decent energy minimization protocol consisting of 30,000 cycles was applied, first to the solvent, then to the solvent and side chains of the capsid system. Upon initiating dynamics, the temperature was increased from 60 K to 310 K over an interval of 5 ns, applying Cartesian restraints of 5 kcal mol$^{-1}$ to the protein backbone. Continuing under isothermal, isobaric conditions, backbone restraints were removed over an interval of 5 ns. The capsid system was equilibrated for 0.1 μs without restraints, during which morphological properties of the capsid relaxed from the crystal structure and stabilized. A production simulation totaling 1 μs was performed employing a time-step of 2 fs, saving coordinate frames at intervals of 20 ps.

Simulations were propagated using the r-RESPA integrator available in NAMD. Long-range electrostatics were split from short-range at a cutoff of 12 Å according to a quintic polynomial splitting function and computed with the PME (particle-mesh Ewald) method, as implemented in NAMD. PME utilized a grid spacing of 2.1 Å and eighth-order interpolation. Full electrostatic evaluations were performed every 4 fs. Temperature regulation was carried out with the Langevin thermostat algorithm in NAMD, employing a damping coefficient of 1 ps$^{-1}$. The Nosé-Hoover Langevin piston control was applied to maintain constant pressure of 1 bar, allowing isotropic cell scaling, with piston oscillation period of 2,000 fs and damping timescale of 1,000 fs. All bonds to hydrogen were constrained with the SHAKE (solute) or SETTLE (solvent) algorithms.

## Morphology characterization

Capsid inner volume and sphericity were estimated based on fitting a polyhedron of 240 triangular faces within the capsid surface (*Figure 1*). The polyhedron was constructed for each simulated conformation by connecting the center points of the groups of five to six Cp chains surrounding each symmetry axis or pore: Fivefold vertices were defined as the center of $A_5$ chains (capsid pentamers). Sixfold vertices were defined as the center of $B_2C_2D_2$ chains (capsid hexamers). Threefold vertices were defined as the center of $B_3C_3$ chains at hexamer interfaces. Points representing the triangular pores at the interfaces of pentamers and hexamers were defined as the center of $A_2B_2C_1D_1$ chains.

The surface area *SA* of the polyhedron was calculated as the sum over the areas of 240 triangular faces, according to *Equation 1*, where $b$ is the base of each triangle and $h$ is the height.

$$SA = \sum_{Triangles}^{240} \frac{bh}{2} \tag{1}$$

The volume $V$ of the polyhedron was calculated as the sum over the volumes of 240 triangular pyramids formed by each triangular face and the capsid center, according to *Equation 2*, where $bh/2$ is the area of the base of each triangular pyramid and $H$ is the pyramid height.

$$V = \sum_{Pyramids}^{240} \frac{\frac{bh}{2}H}{3} \tag{2}$$

The sphericity $\Psi$ of the polyhedron was calculated as a ratio of volume to surface area, according to *Equation 3* (*Wadell, 1935*). This formula gives rise to unit sphericity for a perfect sphere and ~0.939 for a regular icosahedron.

$$\Psi = \frac{\pi^{\frac{1}{3}}(6V)^{\frac{2}{3}}}{SA} \tag{3}$$

## Spike fluctuation analysis

The crystallographic version of AB and CD dimers (*Bourne et al., 2006*) were used as reference structures for alignment. Four points were defined within the reference structures based on the geometric center of Cαs of selected residues: dimer core (residues 50 to 54 and 104 to 108), dimer spike tip (residues 76 to 80), and C-terminus of helix 5 for chain A/C and chain B/D (residues 122 to 126). The point representing the dimer core was set to the origin. A vector fitted through points representing the dimer core and spike tip was aligned along the $z$-axis. A vector fitted through points representing the C-terminus of helix 5 for A and B or C and D chains, respectively, was aligned along

the *x*-axis (*Figure 2d–e*). Dimer conformations extracted from the capsid simulation were aligned to their appropriate reference structures based on their Cα trace. Points representing dimer spike tips were determined for each simulated dimer conformation and reported as instantaneous position in the *xy* plane.

## Principal component analysis

PCA was performed as described by (*Perilla and Schulten, 2017*). PCA visualization employed the NMWiz plugin (*Bakan et al., 2011*) in VMD.

## Theoretical resolution experiments

A subset of 1,000 frames was extracted at 1-ns intervals over the final 1 μs of simulation. A steepest decent energy minimization protocol consisting of 30,000 cycles was applied to each frame, such that the capsid conformers contained within were subsequently treated as analogous to distinct capsid particles flash-frozen in vitreous ice. Conformers were aligned on their Cα trace for structural analysis. Theoretical density maps were rendered for each conformer at an arbitrary resolution and pixel size using the pdb2mrc function from EMAN2 (*Tang et al., 2007*), effectively creating a Gaussian density distribution for each atom with 1/e width at 1/resolution. Icosahedral averaging and FSC functions were also supplied by EMAN2. For overall resolution estimation, the maps were randomly shuffled to eliminate time-ordering, divided into two subsets and averaged, and FSC calculations were performed on the two half-maps. For local resolution estimation, as well as assignment of local resolution values to atomic coordinates, the LocalFSC plugin (*Cardone et al., 2013*) from UCSF Chimera (*Pettersen et al., 2004*) was used with an adaptive box size. The local resolution is plotted as the average over the four chains in the capsid asymmetric unit.

## Solvent exchange rate analysis

(*Perilla and Schulten, 2017*) developed a 3D ray-tracing method within the framework of VMD to distinguish the interior versus exterior of the HIV-1 capsid for the purpose of measuring solvent exchange rates. A similar method was employed here for the HBV capsid, with some modifications. Instead of casting six rays in the $\pm\hat{x}$, $\pm\hat{y}$, and $\pm\hat{z}$ directions, a total of 512 rays were cast based on Poisson disk sampling (*Bridson, 2007*) to achieve improved distribution of ray strikes along the continuously convex inner surface of the spherical capsid. Exchange rates were calculated based on the numbers of solvent species that were present on one side of the capsid surface at a reference frame $t_0$ and then found on the opposite side of the capsid surface at a subsequent frame $t > t_0$ (*Andoh et al., 2014*; *Perilla and Schulten, 2017*). The slope of the linear fit to cumulative number of species versus simulation time segment gives the respective rate. Exchange rates for solvent crossing a spherical surface with the same average radius as the capsid (145.4 Å) were estimated based on a 4.4-ns simulation of a 392 Å³ box of explicit solvent (150 mM NaCl), performed using the same parameters and protocols described for simulation of the capsid.

## Ion localization analysis

Ion occupancy maps were calculated at a grid resolution of 1 Å for the final 1 μs of simulation, averaging over 50,000 frames using the volmap plugin in VMD. The volutil plugin in VMD was employed to convert occupancy to a potential of mean force representing Gibbs free energy of binding (ΔG) based on *Equation 4*, where $\frac{\rho(\vec{r})}{\rho_0}$ is the ratio of species occupancy at a given site (grid voxel) versus its occupancy in bulk solvent (explained in detail by [*Cohen et al., 2006*]).

$$W(\vec{r}) = -k_B T \ln \frac{\rho(\vec{r})}{\rho_0} \tag{4}$$

Global ion localization was evaluated based on a single average occupancy map encompassing the entire system, aligned on the capsid's Cα trace. Bulk occupancy values of $\rho_0 = 0.003$ and 0.002, taken from a bulk solvent region at the capsid center, were employed for conversion to ΔG for sodium and chloride ions, respectively. Ion localization around the capsid asymmetric unit was evaluated based on a map obtained from averaging over all 60 copies of the asymmetric unit present in the capsid assembly, aligned on Cα trace. This approach provided a total of 60 μs of effective

system sampling. Bulk occupancy values of $\rho_0$ = 0.003 and 0.002, taken from the bulk region of the overlay, were employed for sodium and chloride ions, respectively. Bulk values are greater for sodium than chloride due to the 1680 additional sodium ions required to neutralize the highly negative capsid structure.

### Data availability

The simulation dataset that supports the findings of this work has a footprint of 4 TB, and is, thus, unavailable for direct download. Instead, the dataset is available from the corresponding author upon request. Interested readers should contact the corresponding author to determine the best approach for managing data transfer between institutions.

## Acknowledgements

This work is dedicated to Klaus Schulten (1947-2016), who pioneered the application of MD simulations to study complete virus capsids. The authors acknowledge funding from the National Institutes of Health grant 5P30GM110758-04 and the University of Delaware. This research is part of the Blue Waters sustained-petascale computing project, which is supported by the National Science Foundation (awards OCI-0725070 and ACI-1238993) and the state of Illinois. Blue Waters is a joint effort of the University of Illinois at Urbana-Champaign and its National Center for Supercomputing Applications. The authors acknowledge use of Blue Waters through an allocation for research undertaken at the University of Illinois at Urbana-Champaign entitled 'Unveiling the functions of the HIV-1 and HBV capsids through the computational microscope.'

## Additional information

### Competing interests

Adam Zlotnick: A.Z. has an interest in a biotechnology company and acknowledges a conflict of interest. The other authors declare that no competing interests exist.

### Funding

| Funder | Grant reference number | Author |
|---|---|---|
| National Institutes of Health | Center of Biomedical Research Excellence Grant,5P30GM110758-04 | Jodi A Hadden Juan R Perilla |
| University of Delaware | Postdoctoral Fellowship | Jodi A Hadden |
| National Institutes of Health | Biomedical Technology Research Resource,9P41GM104601 | Klaus Schulten |

The funders had no role in study design, data collection and interpretation, or the decision to submit the work for publication.

### Author contributions

Jodi A Hadden, Conceptualization, Resources, Data curation, Software, Formal analysis, Supervision, Funding acquisition, Validation, Investigation, Visualization, Methodology, Writing—original draft, Project administration, Writing—review and editing; Juan R Perilla, Conceptualization, Resources, Software, Formal analysis, Supervision, Funding acquisition, Validation, Visualization, Methodology; Christopher John Schlicksup, Conceptualization, Formal analysis, Validation, Visualization, Methodology, Writing—review and editing; Balasubramanian Venkatakrishnan, Conceptualization, Formal analysis, Validation, Methodology; Adam Zlotnick, Conceptualization, Resources, Supervision, Methodology, Project administration, Writing—review and editing; Klaus Schulten, Resources, Software, Supervision, Funding acquisition, Methodology, Project administration

### Author ORCIDs

Jodi A Hadden (iD) http://orcid.org/0000-0003-4685-8291
Juan R Perilla (iD) http://orcid.org/0000-0003-1171-6816

Adam Zlotnick (iD) http://orcid.org/0000-0001-9945-6267
Klaus Schulten (iD) https://orcid.org/0000-0001-7192-9632

**Decision letter and Author response**
Decision letter https://doi.org/10.7554/eLife.32478.027
Author response https://doi.org/10.7554/eLife.32478.028

## Additional files

**Supplementary files**
• Transparent reporting form
DOI: https://doi.org/10.7554/eLife.32478.025

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
