## [Decision Letter]

Thank you for submitting your article "Chemical-physical characterization of the hepatitis B virus capsid by all-atom molecular dynamics simulations" for consideration by *eLife*. Your article has been reviewed by three peer reviewers, and the evaluation has been overseen by a Reviewing Editor and Wenhui Li as the Senior Editor. The following individual involved in review of your submission has agreed to reveal his identity: William Gelbart (Reviewer #3).

The reviewers have discussed the reviews with one another and the Reviewing Editor has drafted this decision. Major revisions are required in order to make this work potentially acceptable for *eLife*. A final decision will be made once we receive your revisions.

Summary:

In this work, the authors use unrestrained molecular dynamics simulations to characterize the structural dynamics of the icosahedral HBV capsid. This study extends both time sampling and level of detail for previous work at the coarse-grained level for the same system, and shows insights into the function of the capsid. In addition, they show that capsid dynamics limits achievable resolution in cryoEM data processing for this system.

Overall, in its current form, the paper reads like a long description of the simulation and its analyses, and the reader may miss the exciting bits along the way, especially if they are not experts in molecular dynamics simulations. The paper should be rearranged to present from the get go the most exciting findings. Even the title and Abstract fail to do so currently. The Results sections should have meaningful section titles announcing the main finding to be expected (not just a narration of the analysis and figures). The separation between the Results section and the Discussion renders the former very dry. I suggest that the main findings and their contextualization with other data is included in the Results, with a shorter Discussion section.

Moreover, the authors should relate their results in more detail to previous findings, and to elaborate on the idea of suggesting treatment for disease. A couple of specific testable predictions would be very useful.

Below are more specific major concerns about the work and the presentation of the results.

Essential revisions:

1) What are the biological / biochemical / biophysical implications for the capsid hexamers to be more flexible than pentamers? (Subsection “Capsid flexibility and dynamics”, fourth paragraph).

2) The determination of larger-scale collective modes still appears to be challenging for this system. The authors present a normal mode analysis in Figure 5, but as they state in the Discussion, the results are difficult to interpret. Have the authors considered a principal component analysis of the capsid dynamics? This could bring to light similar motions but with more clarity.

3) The exchange rates for the water molecules (subsection “Solvent exchange across the capsid”) of 4.7x10^6 / ns seem extremely high. Do these rates imply that there are a lot of water molecules moving in/out per nanosecond? Are these waters bouncing back and forth? How does this compare to other viruses (or to bulk water, if one was to define a surface)? The rates for the sodium ions are more reasonable. The authors should also show the channels or paths through which the water molecules flow. Are there channels that lead water molecules in, and others out? Or is the exchange happening both ways in a single channel? It could be interesting to know more of the detailed structure on what is happening with this process. This is something interesting that can only be seen or suggested based on their heroic calculations.

4) Regarding the ion location around the capsid, Figure 7A is not really clear enough to see the positioning of Asp78 and Glu77; could this be improved? Also, I do not understand the delta*G* values presented along each panel in Figure 7. What are these values exactly? Is it the density / location of sodium ions at different delta-*G* cutoffs? More details should be given here.

5) It is questionable if molecular simulation / free energy calculations would yield meaningful delta-delta*G*'s even, for a range so small (within 1 kcal / mol over the set of 4 panels). Perhaps the authors could remove some of the cutoff analysis and instead focus on rationalizing the mechanism of action for the CpAM's (allosteric modulators), which somehow appear to be interacting with or modulated by the ion binding locations.

6) In the second paragraph of the subsection “Capsid flexibility and dynamics”, the authors refer to promoter chains C and D, but this is not introduced until Figure 3 (although they refer her only to Figure 2).

7) In Figure 3, it seems like the color scheme of the angles is related to the symmetry points, which is not the case. Also, the plots are a deviation from the angle, so they should be labeled deltaϴ.

8) Figure 4 seems to cover only (the last?) 1us of simulation (but says it's 1.1us). It is not evident at all to me that panels B and C show more symmetry breaking than others, e.g., why is C more dramatic than E?

9) In Figure 8—figure supplement 1, the authors should specify a little better what was done. The sentence: “Two average maps (…) were rendered to 1 °A resolution, with a pixel size of 0.25 °A.” Does this mean that they were low-pass filtered to 1 A (this would be better wording than "rendered to 1A") but then the correlation between the two maps was only 2.3? In that figure, the actual values of the FSC at 0.143 should be indicated for all curves (in A). Given the modes are not correlated, why would the authors think local resolution would be a valuable metric to use? Also, for panel D, it seems that they altered the sampling rate (pixel size), so two things are changing here, but no comment is done on either of them (whether there were expectations, etc.).

Essential revisions of the presentation:

10) Please simplify the rather extensive analysis into more digestible findings. Some figures could be moved to Supplementary Information (e.g. Figure 3) to help focus the big take-home messages. Along the same lines, Figure 1A should have a scale bar and the authors should discuss in the introduction how many MDa and/or atoms are in the system to quickly characterize the size of the capsid for those less familiar.

11) The findings relating to cryo-EM are interesting, they probably should not be the main message of the paper. In this part of their analysis, the authors should emphasize that in this new era of increasing resolution in cryo-EM, capsid flexibility might become the rate-limiting factor to achieving true atomic (1-2 A) resolution, and propose that symmetry relaxation methods are attempted.

12) While some references are cited (e.g., Tama and Brooks, May and Brooks), it would be helpful to have some discussion in the text of theoretical work that has been done on the collective (e.g., normal mode) motions of viral capsids and the extent to which the present all-atom simulations confirm and extend these earlier analyses.

13) There appear to be no references to recent work being done on asymmetric reconstructions of viral capsids [see, for example, the work on bacteriophage MS2: X. Dai, Z. Li, M. Lai, S. Shu, Y. Du, Z.H. Zhou and R. Sun. In situ structures of the genome and genome-delivery apparatus in a single-stranded RNA virus. Nature, 541 (7635), 112-116 (2017)]. It would be helpful to call the reader's attention to the fact that cryo-EM reconstructions of this kind have been carried out in which icosahedral symmetry is not assumed/imposed. Indeed, this new work is what makes the present manuscript even more interesting, and vice versa.

14) The title should capture the big discovery presented in the paper, to help funnel the readers to the findings and conclusions. It might be helpful to include explicit mention of the importance of dynamics and asymmetry. E.g., "Chemical-physical characterization of dynamics and asymmetry in the hepatitis B virus capsid, as determined by all -atom molecular dynamics simulations", or just "Dynamics and asymmetry in the hepatitis B virus capsid, as determined by all -atom molecular dynamics simulations".

Optional revision:

15) PCA analysis on the asymmetric unit or a subset of the entire system should be performed. Such an analysis may reveal clearer modes that could then be used as reduced variables back into the large simulation to find simpler motion correlations. For instance, perhaps showing the motions of the virus only as related to the previously suggested motions (expansion, twisting of Cps, etc.) and showing that they are not occurring (or at least not alone) may be informative.

---

## [Author Response]

Summary:In this work, the authors use unrestrained molecular dynamics simulations to characterize the structural dynamics of the icosahedral HBV capsid. This study extends both time sampling and level of detail for previous work at the coarse-grained level for the same system, and shows insights into the function of the capsid. In addition, they show that capsid dynamics limits achievable resolution in cryoEM data processing for this system.Overall, in its current form, the paper reads like a long description of the simulation and its analyses, and the reader may miss the exciting bits along the way, especially if they are not experts in molecular dynamics simulations. The paper should be rearranged to present from the get go the most exciting findings. Even the title and Abstract fail to do so currently. The Results sections should have meaningful section titles announcing the main finding to be expected (not just a narration of the analysis and figures). The separation between the Results section and the Discussion renders the former very dry. I suggest that the main findings and their contextualization with other data is included in the Results, with a shorter Discussion section.

We are very grateful to the reviewers for supporting revision of the manuscript. We have completely re-written and rearranged the text to ensure the manuscript now emphasizes discoveries instead of descriptions. The manuscript includes a new title and Abstract. The Results sections now open with headings that describe the major findings and present from the get-go the important outcomes from analyses. Results are now also contextualized with previously reported data and important commentary. Discussion now only synthesizes information from multiple analyses and provides speculation on what results could mean with regard to biological function of the capsid.

Moreover, the authors should relate their results in more detail to previous findings, and to elaborate on the idea of suggesting treatment for disease. A couple of specific testable predictions would be very useful.

We thank the reviewers for further guiding us to improve the manuscript. We have now included additional discussion in each section relating our results to previously reported data characterizing capsid dynamics and capsid influence on solvent environment, particularly from other MD simulation studies. We have suggested insights into biological function of the capsid based on our results, and elaborated on treatments that could be designed to inhibit these functions.

Below are more specific major concerns about the work and the presentation of the results.Essential revisions:1) What are the biological / biochemical / biophysical implications for the capsid hexamers to be more flexible than pentamers? (Subsection “Capsid flexibility and dynamics”, fourth paragraph).

We thank the reviewers for encouraging us to relate all observations to biological function to extract the most meaning from our results. On this point, we have endeavored to be more specific. Increased flexibility of hexamers actually stems from increased flexibility of CD dimers, particularly in the spike tip. We have created a separate Results section that includes additional analyses to elaborate on dimer spike flexibility and how it is altered by quasi-equivalence. We have included new comments in the manuscript describing our thoughts on this finding, as well as a hypothesis for how spike flexibility, particularly the flexibility of CD dimer spikes, could play a role in the capsid’s cellular signaling process.

2) The determination of larger-scale collective modes still appears to be challenging for this system. The authors present a normal mode analysis in Figure 5, but as they state in the Discussion, the results are difficult to interpret. Have the authors considered a principal component analysis of the capsid dynamics? This could bring to light similar motions but with more clarity.

We thank the reviewers for their great suggestion and for bringing to light our failure to describe the approach clearly in the text. The analysis presented in the original version of the manuscript was not normal modes analysis, but was already principal component analysis. We have re-written the section to clarify our approach and included more discussion regarding the observation of asymmetry in previous studies of icosahedral capsids. The manuscript now includes commentary regarding the possibility that the symmetric features of HBV capsid modes may be more pronounced once more extensive conformational sampling of the capsid is available. Given the difficulty in interpreting the results of this section, we only draw conclusions based on global capsid motions apparent in the lowest frequency modes.

3) The exchange rates for the water molecules (subsection “Solvent exchange across the capsid”) of 4.7x10^6 / ns seem extremely high. Do these rates imply that there are a lot of water molecules moving in/out per nanosecond? Are these waters bouncing back and forth? How does this compare to other viruses (or to bulk water, if one was to define a surface)? The rates for the sodium ions are more reasonable. The authors should also show the channels or paths through which the water molecules flow. Are there channels that lead water molecules in, and others out? Or is the exchange happening both ways in a single channel? It could be interesting to know more of the detailed structure on what is happening with this process. This is something interesting that can only be seen or suggested based on their heroic calculations.

We are extremely grateful to the reviewers for catching our mistake. The exchange rates for water should be on the order of 10^3, not 10^6, and we have corrected this typo. The reviewers are correct in understanding that these are the numbers of waters moving in and out of the capsid per nanosecond (4.7x10^3 water molecules moving in per nanosecond and 4.7x10^3 waters moving out per nanosecond).

As we have now made clear in the text, the solvent exchange analysis represents the transition of water molecules from bulk solvent inside to bulk solvent outside the capsid and vice versa. Boundary waters close to the protein surface are neglected, so the water molecules included in the reported exchange rates are not simply bouncing back and forth. We have added comments in the manuscript noting that solvent exchange across the capsid surface occurs both inward and outward through all pore types.

We have augmented the discussion of these exchange rates with comparisons to solvent exchange rates measured for other virus capsids (HIV-1, PCV2, and poliovirus), and have performed an additional analysis to compare our solvent exchange rates to that of bulk water crossing a spherical surface – we thank the reviewers for this great suggestion.

4) Regarding the ion location around the capsid, Figure 7A is not really clear enough to see the positioning of Asp78 and Glu77; could this be improved? Also, I do not understand the deltaG values presented along each panel in Figure 7. What are these values exactly? Is it the density / location of sodium ions at different deltaG cutoffs? More details should be given here.

We thank the reviewers for guiding us in improving the clarity of our Results presentation. We have doubled the size of the indicated figure panel so that sodium localization around spike tips is more visible. With regard to deltaG values, we have provided additional explanation in the text describing how potentials of mean force for free energy of binding can be calculated from occupancy maps. The mathematical procedure we use is also detailed in Materials and methods, and readers are referred to a reference for further information.

5) It is questionable if molecular simulation / free energy calculations would yield meaningful delta-deltaG's even, for a range so small (within 1 kcal / mol over the set of 4 panels). Perhaps the authors could remove some of the cutoff analysis and instead focus on rationalizing the mechanism of action for the CpAM's (allosteric modulators), which somehow appear to be interacting with or modulated by the ion binding locations.

We thank the reviewers for their great suggestions and for pointing out instances where our presentation was not clear. We have eliminated focus on deltaG cutoff analysis and removed a non-essential figure panel from the ion occupancy discussion that contributed to its emphasis. We have also removed indication of CpAM binding sites in the figure illustrating sodium binding, as upon further consideration, it was very misleading. The sodium binding sites are not close enough to the CpAM binding sites to cause us to suspect that they interact at all, although the orientation of the original figure could suggest this. We have altogether removed mention of CpAMs in this section so that the Results presentation remains focused and clear.

6) In the second paragraph of the subsection “Capsid flexibility and dynamics”, the authors refer to promoter chains C and D, but this is not introduced until Figure 3 (although they refer her only to Figure 2).

We are grateful to the reviewers for pointing out our oversight. We have rearranged the Results presentation and the order of figures in the manuscript so that this problem no longer persists.

7) In Figure 3, it seems like the color scheme of the angles is related to the symmetry points, which is not the case. Also, the plots are a deviation from the angle, so they should be labeled deltaϴ.

We thank the reviewers for their keen observations. We have corrected both issues pointed out for this figure, reduced its complexity, and moved it to the supplement.

8) Figure 4 seems to cover only (the last?) 1us of simulation (but says it's 1.1us). It is not evident at all to me that panels B and C show more symmetry breaking than others, e.g., why is C more dramatic than E?

We are grateful to the reviewers for reading our manuscript so carefully. We have addressed the reported issue with the figure legend. Also, we agree with the reviewers that all figure panels show significant symmetry breaking, and we have thus removed unnecessary emphasis on specific panels.

9) In Figure 8—figure supplement 1, the authors should specify a little better what was done. The sentence: “Two average maps (…) were rendered to 1 °A resolution, with a pixel size of 0.25 °A.” Does this mean that they were low-pass filtered to 1 A (this would be better wording than "rendered to 1A") but then the correlation between the two maps was only 2.3? In that figure, the actual values of the FSC at 0.143 should be indicated for all curves (in A). Given the modes are not correlated, why would the authors think local resolution would be a valuable metric to use? Also, for panel D, it seems that they altered the sampling rate (pixel size), so two things are changing here, but no comment is done on either of them (whether there were expectations, etc.).

We thank the reviewers again for guiding us in improving the clarity of our manuscript. We have re-written this figure legend, as well as revised the explanation of our approach in the text. Instead of describing maps as being rendered at a certain resolution, we provide a more accurate description: “For density map calculation, the individual atoms were treated as Gaussian density distributions with a width (i.e. diameter) of 1 Å at half maximal density.” We have also now indicated the values of the FSC curves at 0.143 in figure panel A, and repeated the analysis from panel D without the variable of pixel size, now eliminating the need for panel D to demonstrate that altering pixel size has no effect).

We apologize to the reviewers if the motivation for local resolution analysis was unclear. Local resolution provides information regarding structural order or relative atomic ﬂuctuation of protein residues. The averaging of density maps calculated from simulated capsid conformations causes resolution to blur, and local resolution analysis relates this blurring to the local motions of protein residues. Resolution analysis is unconcerned with correlated motions within capsid dynamics and is, thus, unrelated to modes from principal component analysis. We determined local resolution from calculated density maps in order to draw comparisons to other measurements of local protein fluctuation, such as crystallographic B-factors and Calpha RMSF determined from MD simulation.

Essential revisions of the presentation:10) Please simplify the rather extensive analysis into more digestible findings. Some figures could be moved to Supplementary Information (e.g. Figure 3) to help focus the big take-home messages. Along the same lines, Figure 1A should have a scale bar and the authors should discuss in the introduction how many MDa and/or atoms are in the system to quickly characterize the size of the capsid for those less familiar.

We are very grateful to the reviewers for guiding us to improve the presentation of our work, and for giving us the chance to significantly revise the manuscript. The original manuscript has now been completely re-written. The Results sections now open with headings describing the main findings and the technical nature of the text has been reduced. The manuscript now focuses on presenting discoveries instead of descriptions. We have taken the reviewers’ advice and moved the original Figure 3 to supplement and added a scale bar to Figure 1A. Both MDa and number of atoms of the system are now clearly indicated early in the text.

11) The findings relating to cryo-EM are interesting, they probably should not be the main message of the paper. In this part of their analysis, the authors should emphasize that in this new era of increasing resolution in cryo-EM, capsid flexibility might become the rate-limiting factor to achieving true atomic (1-2 A) resolution, and propose that symmetry relaxation methods are attempted.

We thank the reviewers for their insightful comments, but feel our findings related to cryo-EM resolution are an important message in the manuscript. An investigation of the role of protein flexibility in limiting cryo-EM resolution is highly relevant to the structural biology field, and comes at a critical time as cryo-EM resolution continues to improve. Our analysis strongly suggests that, regardless of further advances in cryo-EM, achievable resolution will eventually hit a wall due to protein flexibility. We have taken the reviewers’ advice and emphasized in the text that flexibility may become a major limiting factor to achieving true atomic resolution, as well as proposed, in light of our findings regarding asymmetry in the capsid, that symmetry relaxation methods may warrant greater consideration in future structural studies.

12) While some references are cited (e.g., Tama and Brooks, May and Brooks), it would be helpful to have some discussion in the text of theoretical work that has been done on the collective (e.g., normal mode) motions of viral capsids and the extent to which the present all-atom simulations confirm and extend these earlier analyses.

We thank the reviewers for their great suggestion to improve the manuscript. We have augmented the principal component analysis section with more extensive discussion of previous work that applied normal modes analysis to icosahedral capsids. While the modes from our study proved difficult to interpret, we have nevertheless provided some comparison to previously computed normal modes for the HBV capsid. Also, we have added discussion regarding the observation of asymmetry in previous normal modes work on icosahedral capsids, as well as the supposed role of such modes in structural transitions important to function, which likely indicates that the asymmetry we observe in our modes is not unreasonable.

13) There appear to be no references to recent work being done on asymmetric reconstructions of viral capsids [see, for example, the work on bacteriophage MS2: X. Dai, Z. Li, M. Lai, S. Shu, Y. Du, Z.H. Zhou and R. Sun. In situ structures of the genome and genome-delivery apparatus in a single-stranded RNA virus. Nature, 541 (7635), 112-116 (2017)]. It would be helpful to call the reader's attention to the fact that cryo-EM reconstructions of this kind have been carried out in which icosahedral symmetry is not assumed/imposed. Indeed, this new work is what makes the present manuscript even more interesting, and vice versa.

We thank the reviewers for another great suggestion. We have added a short discussion of these and other asymmetric reconstructions available in the literature and contextualized our theoretical image reconstruction within the framework of these previous experimental works.

14) The title should capture the big discovery presented in the paper, to help funnel the readers to the findings and conclusions. It might be helpful to include explicit mention of the importance of dynamics and asymmetry. E.g., "Chemical-physical characterization of dynamics and asymmetry in the hepatitis B virus capsid, as determined by all -atom molecular dynamics simulations", or just "Dynamics and asymmetry in the hepatitis B virus capsid, as determined by all -atom molecular dynamics simulations".

We thank the reviewers for guiding us to select a title that funnels readers to our work. The manuscript title now encompasses the discoveries presented in the paper: “All-atom molecular dynamics simulations of the hepatitis virus capsid reveal insights into biological function and cryo-EM resolution limits.”